# ReDiTT: Retrieval Augmented Conditional Diffusion Transformers for Asynchronous Time Series

**Saiyue Lyu**[†]                                                    *saiyue.lyu@ubc.ca*
*Department of Computer Science, University of British Columbia*

**Zhitian Zhang**                                                    *andy.zhang@rbc.com*
*RBC Borealis*

**Ruizhi Deng**                                                    *ruizhi.deng@borealisai.com*
*RBC Borealis*

**Thibaut Durand**                                                    *thibaut.durand@borealisai.com*
*RBC Borealis*

**Reviewed on OpenReview:** *https://openreview.net/forum?id=sv35KiCipb*

## Abstract

We present a diffusion based model for asynchronous time series prediction, where the goal is to predict the next inter event time and event type. To address the inherent uncertainty of future events, we introduce ReDiTT, a retrieval augmented conditional diffusion transformer that operates in latent space. ReDiTT retrieves structurally similar latent sequences from a memory bank during both training and inference and incorporates them as reference conditions through cross attention. This retrieval based conditioning allows the model to attend to relevant temporal dynamics and provides global structural guidance for generation. As a result, ReDiTT stabilizes long horizon forecasting and improves sample diversity. Experiments on seven real world datasets demonstrate state of the art performance on next event prediction and long horizon forecasting. Our code is available at https://github.com/BorealisAI/ReDiTT.

## 1 Introduction

Asynchronous time series (*a.k.a.* continuous-time event sequence) prediction arises in a wide range of real world applications, including event driven systems (Enguehard et al., 2020), healthcare monitoring (Lorch et al., 2018; Rizoiu et al., 2018), finance (Bacry et al., 2015; Jin et al., 2020), and user behavior modeling (Hernandez et al., 2017; Zhang et al., 2022; Kong et al., 2023), where observations occur at irregular time intervals rather than on a fixed grid. Unlike regularly sampled time series, asynchronous data encodes information jointly in both event values and inter-event times, leading to complex temporal dynamics that are highly stochastic and nonstationary (Xue et al., 2023). Accurately modeling such data is crucial for downstream tasks such as forecasting, simulation, and decision making, yet remains challenging due to the sparsity, irregularity, and long-range temporal dependencies inherent in these processes (Schirmer et al., 2022; Zhang et al., 2024).

Asynchronous time series prediction is further complicated by the need to model uncertainty and multimodality over future events, particularly in long-horizon forecasting. Classical autoregressive and likelihood-based temporal point process models often rely on strong parametric assumptions or Markovian dynamics (Hawkes, 1971; Mei & Eisner, 2017; Zhang et al., 2020; Zuo et al., 2020; Yang et al., 2021), limiting their ability to

---

[†]Corresponding author. Work done during an internship at RBC Borealis.

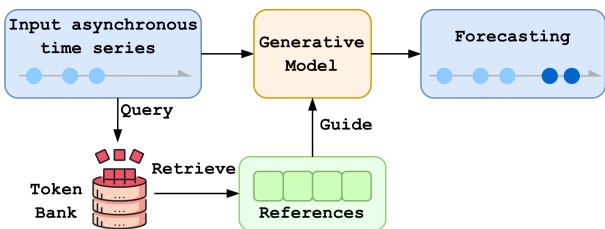

Figure 1: Overview of ReDiTT framework. Given an observed event history, we retrieve the similar sequences from a token memory bank. The retrieved references are injected into diffusion transformer to guide denoising for future forecasting.

generalize to more complex asynchronous time series, particularly those exhibiting rich global structure, long-range dependencies, or mixed continuous and discrete observations.

Recent progress has shown that generative modeling in latent space via variational autoencoders (VAEs) (Higgins et al., 2017) combined with diffusion models (Peebles & Xie, 2023), can effectively capture the stochastic structure of asynchronous time series. In particular, Mukherjee et al. (2025) demonstrates that a VAE can learn a compact and expressive latent representation that supports both accurate reconstruction and diffusion-based next-event and long-horizon forecasting. While diffusion models are powerful generative models, their application to asynchronous time series remains underexplored, despite their ability to generate sequences holistically and mitigate error accumulation compared to autoregressive approaches.

Despite recent progress, unconditional or weakly conditional diffusion models still face significant challenges in long-horizon time series prediction (Liu et al., 2024). Unlike image diffusion models, time series data usually do not come with explicit semantic labels or other strong sources of supervision, which limits the availability of informative global conditioning signals during generation. Without such guidance, the model has less ability to preserve the specific dynamics of an individual trajectory over extended horizons. As a result, long-term forecasts often drift toward generic, high-probability patterns from the training distribution, rather than maintaining trajectory-specific temporal structure and fine-grained event evolution.

To address these limitations, we propose **ReDiTT** : **Re**trieval Augmented Conditional **Di**ffusion **T**ransformers for Asynchronous **T**ime Series. During training, each sequence retrieves its top $k$ nearest neighbors from a token memory bank, which are then used as conditions. These retrieved sequences share the same latent format as the input and are incorporated through cross-attention modules within the diffusion transformer blocks, allowing the model to explicitly attend to structurally similar temporal dynamics. At inference time, we retrieve the top $k$ references from the token memory bank built from training set to guide generation. An overview of ReDiTT is illustrated in Figure 1. This retrieval-augmented conditioning provides global structural guidance that stabilizes long-horizon forecasting, and improves sample diversity by anchoring generation to concrete examples of temporal dynamics rather than relying purely on learned parameters. Our main contributions are:

① We introduce ReDiTT, the first retrieval-based diffusion framework for asynchronous time series prediction that conditions on top-$k$ retrieved latent priors from a pre-constructed latent token bank.

② We propose a novel conditioning approach for retrieve-based diffusion transformer and show it effectively integrates prior information and provides reasonable guidance.

③ We demonstrate that ReDiTT significantly improves next-event and long-horizon prediction with state-of-the-art performances by experiments on seven real world datasets with a comprehensive analysis.

## 2 Related Work

### 2.1 Temporal Point Processes (TPPs)

Marked TPPs are widely adopted as a standard approach for modeling asynchronous time series. A TPP is a stochastic process that generates sequences of discrete events over time. A sequence of $n$ events can be represented as $\mathbf{x} = \{\mathbf{x}_1, \cdots, \mathbf{x}_n\}$, where each event $\mathbf{x}_i = (t_i, e_i)$ consists of the index $i$ indicating the chronological order of events, the inter-event time $t_i$ since the previous event and the corresponding event type $e_i$. Another widely used variant is to represent events by their inter-event intervals $\tau_i = t_i - t_{i-1}$ rather than the absolute timestamps $t_i$. These two parameterizations are essentially equivalent, so the literature often switches between them without loss of generality. Classical TPP models (Mei & Eisner, 2017) parameterize conditional intensity functions for next-event prediction and are trained by maximizing the log-likelihood of observed event sequences. Prior work on temporal point processes has largely focused on neural architectures that extend likelihood-based intensity modeling, beginning with RNN-based approaches (Du et al., 2016; Mei & Eisner, 2017) and later incorporating more expressive designs to better capture uncertainty (Mehrasa et al., 2019; Lüdke et al., 2023). More recent Transformer-based TPPs leverage attention mechanisms to improve long range dependency modeling (Zhang et al., 2020; Zuo et al., 2020; Yang et al., 2021), but still rely on autoregressive intensity formulations. However, the goal has extended beyond forecasting the immediate next event to generating the full future event trajectory, and classical and neural TPPs can be unsatisfactory since errors introduced at early steps accumulate as the autoregressive rollout proceeds. On the optimization side, likelihood-based training typically requires computing distributional quantities implied by the learned intensity, which can become expensive.

### 2.2 Diffusion Models for Asynchronous Time Series

The shortcomings of classic neural TPP approaches motivate diffusion and flow matching approaches. These methods treat forecasting as conditional generation, transforming simple base noise into future event sequences given the history. By modeling the joint continuation rather than repeatedly sampling one event at a time, diffusion based approaches can better support long-horizon generation and produces diverse futures for uncertainty quantification, offering a strong non-autoregressive alternative to intensity-based neural TPPs. Early diffusion work in time series focused on learning conditional distributions for tasks like imputation (Tashiro et al., 2021) and probabilistic forecasting (Rasul et al., 2021), demonstrating that iterative denoising can capture uncertainty without being constrained to a single greedy rollout. For temporal point processes specifically, Add and Thin (Lüdke et al., 2023) formulates diffusion over full marked event sequences and outperforms autoregressive TPP forecasters. Zhou et al. (2025) embed event sequences into a vector space and diffuse to forecast the full future sequence in one go. Yuan et al. (2023) extended diffusion to spatio-temporal setting. Zeng et al. (2024) coupled two diffusions, one for inter-arrival times and one for event types, to model the joint distribution. More recently, ADiff4TPP (Mukherjee et al., 2025) proposed an asynchronous noise schedule in a VAE latent space to strengthen conditioning via earlier event history when predicting further into the future. While promising, these approaches yield only limited empirical gains, and diffusion for asynchronous event modeling is still relatively understudied. We propose retrieval based diffusion transformers to tackle these challenges and better predict long-horizon future.

### 2.3 Retrieval Augmented Generation

Retrieval has become a practical way to enhance generative models with non-parametric memory. For text, $k$NN language models (Khandelwal et al., 2019) augment a base LM by retrieving semantically similar contexts from a data store at inference time. For image, retrieval-augmented diffusion models (Blattmann et al., 2022) condition denoising on retrieved reference images to better capture specific visual structure and long-tail concepts. In time series forecasting, several recent methods (Han et al., 2025; Ning et al., 2025; Tire et al., 2024; Li, 2025) leverage retrieval by selecting historically similar segments and using their continuations as additional context. RATD (Liu et al., 2024) further combines retrieval with diffusion by injecting retrieved references into the denoising process. However, these approaches are primarily designed for regularly sampled time series segments and do not directly apply to asynchronous marked event streams

without nontrivial changes. In contrast, our method is tailored to event prediction: we perform retrieval in a VAE latent space that preserves event wise structure, and condition a latent diffusion transformer on retrieved references represented in the same event-sequence format. This design enables trajectory-specific guidance for coherent long horizon event generation, rather than serving only as a segment level short range forecasting booster.

## 3 Preliminary

### 3.1 Asynchronous Time Series Forecasting Tasks

Asynchronous time series forecasting is typically evaluated using two tasks. Suppose we observe an event history $\{\mathbf{x}_1, \cdots, \mathbf{x}_i\}$. **Next event prediction** requires a model to forecast the immediate next event $\hat{\mathbf{x}}_{i+1} = (\hat{t}_{i+1}, \hat{e}_{i+1})$, including both its occurrence time or inter-arrival time and its type or mark, conditioned on the observed history. **Long horizon prediction** extends this setting by requiring the model to generate a sequence of future events $\{\hat{\mathbf{x}}_{i+1}, \cdots, \hat{\mathbf{x}}_{i+m}\}$ over a prediction window of length $m$. This task evaluates the model's ability to capture how uncertainty accumulates as the forecasting horizon increases.

### 3.2 Flow Matching for Asynchronous Time Series

Flow matching (FM) (Lipman et al., 2023; Liu et al., 2023; Lee et al., 2024; Esser et al., 2024) provides a diffusion style way to train continuous time generative models by directly regressing a velocity field instead of learning scores. Concretely, FM views the generation as solving an ODE that transports a simple base distribution (typically Gaussian noise) to the data distribution via a learned vector field $\frac{d\mathbf{z}_s}{ds} = v_\theta(\mathbf{z}_s, s)$, using an explicit interpolation path $\mathbf{z}_s = a_s \mathbf{z} + b_s \boldsymbol{\epsilon}$, where $s$ denotes a random time $s \sim \mathcal{U}(0, 1)$ (we use $s$ to denote the timestep in diffusion model, and $t$ to represent the time value for asynchronous time series data), $\mathbf{z}$ is the input clean sample, and $\boldsymbol{\epsilon} \sim \mathcal{N}(0, 1)$ is the Gaussian noise. Rectified flow (Liu et al., 2023; Lee et al., 2024) defines a simpler interpolation $\mathbf{z}_s = (1 - s)\mathbf{z} + s\boldsymbol{\epsilon}$. Conditional Flow Matching (CFM) makes training simulation-free by deriving a direct closed-form conditional vector field $u_s(\mathbf{z}_s|\boldsymbol{\epsilon})$ for this path, and learning $v_\theta$ by a simple regression loss $\mathbb{E}_{s,\boldsymbol{\epsilon},\mathbf{z}}\big[\|v_\theta(\mathbf{z}_s, s) - u_s(\mathbf{z}_s|\boldsymbol{\epsilon})\|^2\big]$.

To facilitate efficient retrieval and ease the diffusion training, each event $\mathbf{x}_i$ is encoded as a compact latent representation $\mathbf{z}_i \in \mathbb{R}^d$ using a pretrained VAE $E_\phi$. Let $\mathbf{x} \in \mathcal{X}$ be an event sequence of length $n$. We pad $\mathbf{x}$ to a fixed length $N$, where $N$ is the maximum sequence length in $\mathcal{X}$. The encoder then maps the padded sequence to a latent representation $\mathbf{z} = E_\phi(\mathbf{x}) \in \mathbb{R}^{N \times d}$, which serves as a single training sample for latent flow matching.

To make flow matching respect the causal and unevenly spaced nature of event sequences, asynchronous matrix valued interpolation (Mukherjee et al., 2025) defines:

$$\mathbf{z}_s = \boldsymbol{A}(s)\mathbf{z} + (\boldsymbol{I} - \boldsymbol{A}(s))\boldsymbol{\epsilon}, \tag{1}$$

where $\boldsymbol{\epsilon} = \{\epsilon_1, \cdots, \epsilon_N\}$ is a Gaussian noise of same dimension $N \times d$ as $\mathbf{z}$. And $\boldsymbol{A}(s) \in \mathbb{R}^{N \times N}$ is a diagonal matrix whose per event schedule is designed so that later events are injected with noise earlier (and therefore are trained to be denoised earlier) than earlier events, that being said, later events are corrupted earlier and the model learns to denoise and forecast the tail under stronger noise.

This yields a generative ODE that incorporates the schedule derivative with the chain rule:

$$\frac{d\mathbf{z}_s}{ds} = \boldsymbol{A}'(s)v_\theta(\mathbf{z}_s, \boldsymbol{A}(s)). \tag{2}$$

In each training iteration, we randomly select a time $s \in [0, 1]$ and generate the corresponding intermediate state $\mathbf{z}_s$ using Equation (1). Training follows a CFM objective in this asynchronous setting by weighting the regression with $\boldsymbol{A}'(s)$:

$$\mathcal{L}_{\mathrm{CFM}}(\theta) = \mathbb{E}_{s,\boldsymbol{\epsilon},\mathbf{z}}\big[\,\|\boldsymbol{A}'(s)\big(v_\theta(\mathbf{z}_s, \boldsymbol{A}(s)) - u_s(\mathbf{z}_s|\boldsymbol{\epsilon})\big)\|^2\,\big]. \tag{3}$$

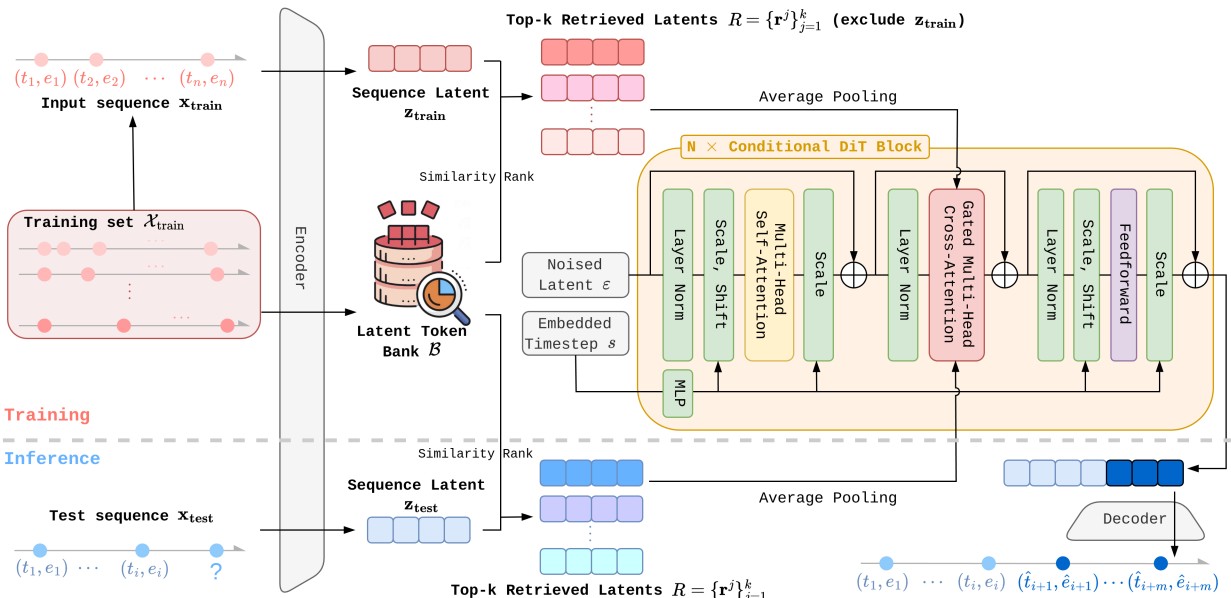

Figure 2: **ReDiTT** models long horizon asynchronous time series forecasting as conditional generation guided by retrieved references. During training, we encode each event sequence $\mathbf{x}_{\text{train}}$ into latent tokens $\mathbf{z}_{\text{train}}$ and build a latent reference bank $\mathcal{B}$. For each sample, we retrieve the top-$k$ nearest neighbors $R$ by latent similarity, aggregate them, and condition a DiT via cross-attention. We then train with the diffusion objective on the future latent segment to denoise and reconstruct clean latents given the prefix and retrieved context. During inference, we encode the observed prefix $\mathbf{x}_{\text{test}} = (t_1, e_1), \ldots, (t_i, e_i)$, apply the same retrieval and aggregation, run the reverse diffusion process to sample future latents, and decode them into event times and types $(\hat{t}_{i+1}, \hat{e}_{i+1}), \cdots, (\hat{t}_{i+m}, \hat{e}_{i+m})$.

During inference, given a observed history $\mathbf{x}_1, \cdots, \mathbf{x}_n$ and a target prediction horizon size $m$. We can form a initial condition $\{\mathbf{z}_1, \cdots, \mathbf{z}_n, \epsilon_{n+1}, \cdots, \epsilon_{n+m}\}$ and we can incorporate the history by filling the corresponding entries of $\frac{d\mathbf{z}_s}{ds}$ in Equation (2) with the known latents:

$$\frac{d\mathbf{z}_s}{ds} = \begin{cases} \boldsymbol{A}'(s)_{ii}(\mathbf{z}_i - \epsilon_i) & \text{if } 1 \le i \le n, \\ \boldsymbol{A}'(s)_{ii}v_\theta(\mathbf{z}_s, \boldsymbol{A}(s)) & \text{if } n < i \le n + m. \end{cases} \tag{4}$$

Standard ODE solvers can solve this without introducing numerical error, which leads us to the predictions $\{\hat{\mathbf{z}}_{n+1}, \cdots, \hat{\mathbf{z}}_{n+m}\}$. The decoder then maps them back to the original space to obtain $\{\hat{\mathbf{x}}_n, \cdots, \hat{\mathbf{x}}_{n+m}\}$.

# 4 Method

To better ground diffusion-based generation in trajectory-specific dynamics, we propose ReDiTT, a retrieval augmented latent Diffusion Transformer for temporal point processes, as illustrated in Figure 2. Given an observed event history $\{\mathbf{x}_1, \cdots, \mathbf{x}_i\}$ as input, our model generates a future event sequence $\{\hat{\mathbf{x}}_{i+1}, \cdots, \hat{\mathbf{x}}_{i+m}\}$, supporting both next event prediction ($m = 1$) and long horizon prediction ($m > 1$). ReDiTT couples latent space embedding retrieval with conditional generation: we retrieve the top-$k$ most similar historical trajectories from a reference bank and use them as additional conditioning to guide the diffusion model towards context consistent futures. We first describe our latent space retrieval strategy in Section 4.1. We then introduce the reference guided conditional flow matching objective in Section 4.2. Finally, we detail the conditional DiT architecture that integrates retrieved references via cross-attention in Section 4.3.

### 4.1 Masked Token Level Retrieval in Latent Space

Directly retrieving asynchronous sequences in the original observation space requires a carefully designed similarity function: each sequence mixes continuous inter-event times with discrete event types, has variable length with padding, and may admit multiple plausible alignments, such as shifted temporal patterns or local event reorderings. Consequently, naive raw-space distances, such as Euclidean distance over concatenated padded features, can be dominated by scale mismatch, sparsity, or padding artifacts, and may not reflect trajectory-level similarity. We therefore perform retrieval in the latent space of the pretrained VAE $E_\phi$, where each event $\mathbf{x}_i = (t_i, e_i)$ is mapped to a compact representation $\mathbf{z}_i \in \mathbb{R}^d$ that is optimized to preserve the information needed for sequence reconstruction. This choice is primarily motivated by representation alignment with the downstream generative model. Since ReDiTT performs diffusion and conditioning over event-sequence latents, latent-space retrieval allows the model to consume retrieved references in the same space as the generation target, without requiring an additional hand-designed metric over mixed continuous-discrete observations. Our raw-space retrieval ablation in Section D.1 shows that carefully normalized raw similarities can also provide useful neighbors, indicating that retrieval-augmented conditioning is robust to the retrieval representation. Nevertheless, latent retrieval offers a simple and model-aligned way to obtain reference trajectories for coherent multi-event forecasting.

Consider an asynchronous sequence $\mathbf{x} = \{\mathbf{x}_1, \cdots, \mathbf{x}_N\}$ with a padding mask $\mathbf{m} \in \{0,1\}^N$ (this mask is common in data processing to keep a constant total event length among all the sequence), we use $E_\phi$ to produce per-event latent tokens $\mathbf{z} = E_\phi(\mathbf{x}) \in \mathbb{R}^{N \times d}$. In addition, we precompute a latent token bank $\mathcal{B} = \{(\mathbf{z} = E_\phi(\mathbf{x}), \mathbf{m}) \,|\, \mathbf{x} \in \mathcal{X}_{\mathrm{train}}\}$ for the training database. We then define a masked cosine similarity between a query $(\boldsymbol{q}, \mathbf{m}^q) \in E_\phi(\mathcal{X})$ and a reference candidate $(\mathbf{r}, \mathbf{m}^r) \in \mathcal{B}$ as a token-wise cosine averaged over valid time steps:

$$\mathtt{sim}(\boldsymbol{q}, \mathbf{r}) = \frac{1}{\sum_i \mathbf{1}[\mathbf{m}_i^q \wedge \mathbf{m}_i^r]} \sum_{i:\mathbf{m}_i^q \wedge \mathbf{m}_i^r} \frac{\langle \boldsymbol{q}_i, \mathbf{r}_i \rangle}{\|\boldsymbol{q}_i\|\|\mathbf{r}_i\|}, \tag{5}$$

where $\frac{\boldsymbol{q}}{\|\boldsymbol{q}\|_2}, \frac{\mathbf{r}}{\|\mathbf{r}\|_2}$ are $\ell_2$ normalized tokens. Finally, we retrieve the top $k$ references for $\boldsymbol{q}$:

$$R(\boldsymbol{q}) = \{\mathbf{r} \,|\, \mathrm{TopK}_{r \in \mathcal{B}} \, \mathtt{sim}(\boldsymbol{q}, \mathbf{r}), \mathbf{r} \neq \boldsymbol{q}\}, \tag{6}$$

which will enter the diffusion process as extra condition. During training, the query sequence is part of the training reference bank, so its nearest neighbor is typically the query sample $\boldsymbol{q}$ itself and would dominate the conditioning signal. Including this exact match would create a degenerate conditioning signal, allowing the model to rely on a duplicate of the input rather than learning to use informative neighboring trajectories. We therefore exclude the closest reference during training and retrieve from the second closest neighbor onward. At inference time, the query comes from the validation or test set while the reference bank contains only training sequences, so the top retrieved neighbor is not an exact duplicate of the query and can be safely used for conditioning.

During evaluation, retrieval follows the natural forecasting setup used for all baselines: only the observed prefix is available when predicting future events. Accordingly, before computing retrieval similarity, we mask all target/future positions in the query sequence and perform retrieval using only the observed prefix. The retrieved sequence itself may contain its full continuation, but it is selected without access to the query's future events. To further examine the effect of this prefix-only retrieval protocol, we include an additional retrieval-only analysis in Section C.2. The results show that prefix-only retrieval remains competitive with full-sequence retrieval on most datasets, indicating that the observed prefix is generally sufficient to retrieve useful neighboring trajectories.

### 4.2 Conditional Flow Matching with Retrieved References

Let $R$ denote the external reference condition associated with a target latent event sequence $\mathbf{z}$. Conditioning does not alter the flow matching construction and Equation (1) still holds, but we now learn a conditional vector field $v_\theta(\cdot \,|\, R)$ that transports samples along this path while being guided by $R$. The conditional flow matching objective becomes:

$$\mathcal{L}_{\mathrm{CFM}}(\theta) = \mathbb{E}_{s,\boldsymbol{\epsilon},\mathbf{z}} \big[ \|\boldsymbol{A}'(s)\big(v_\theta(\mathbf{z}_s, \boldsymbol{A}(s) \,|\, R) - u_s(\mathbf{z}_s | \boldsymbol{\epsilon})\big)\|^2 \big]. \tag{7}$$

Intuitively, $R$ does not modify the geometry of the bridge from $\epsilon$ to $\mathbf{z}$, but it provides additional information that biases the learned dynamics toward trajectory-consistent solutions, which is especially helpful for long-horizon generation.

During training, we retrieve $k$ references $R$ from the same training bank $\mathcal{B}$ and feed the latent tokens as an addition condition in every DiT block, encouraging the conditional flow to model a set of plausible futures consistent with similar historical trajectories. During inference, we retrieve from the same training bank again , so the sequences that model has already encountered can serve as an in-distribution prototype that stabilizes long-horizon generation and reduces drift when extrapolating far beyond the local context.

### 4.3 Conditional Diffusion Transformer Architecture

We modify the Transformer based DiT blocks because they have the following issues when applying to asynchronous time series: conditioning is often injected through adaptive normalization (e.g., AdaLN) where a global conditioning vector (typically built from the diffusion timestep and a sparse label) modulates each block. For asynchronous sequences, this kind of conditioning is intrinsically limited: the timestep carries no instance-specific semantics, and event "labels" (if any) are usually coarse or unavailable, so the conditioning signal is weak and globally broadcast to all tokens in the same way.

After retrieving $k$ nearest-neighbor reference sequences in latent space, we therefore construct a sequence-shaped condition that mirrors the input format. Say we have retrieve $R = \{\mathbf{r}^j\}_{j=1}^k$ for input latent $\mathbf{z}$, we first obtain a reference sequence by weighted pooling:

$$\mathbf{r} = \sum_{j=1}^{k} w_j \mathbf{r}^j \in \mathbb{R}^{N \times d}, \ w_j = \texttt{softmax}\big(\texttt{sim}(\mathbf{z}, \mathbf{r}^j)\big). \tag{8}$$

Before entering the DiT blocks, both current latent input and retrieved reference are expanded to hidden size $D$ by channel repetition, for $\mathbf{r}$, we also broadcast the mask $\mathbf{m^z}$ across the channel dimension. Then $\mathbf{z}$ and $\mathbf{r}$ are augmented with fixed positional embeddings $\texttt{pos}$ to get $\mathbf{z} = \texttt{Repeat}(\mathbf{z}) + \texttt{pos}_\mathbf{z} \in \mathbb{R}^{N \times D}$ and $\mathbf{r} = \texttt{Repeat}(\mathbf{r}) + \texttt{pos}_\mathbf{r} \in \mathbb{R}^{N \times D}$. Considering the reference has the same format as the input, it is naturally to inject $\mathbf{r}$ via cross-attention after self attention in each DiT block as illustrated in Figure 2. After the LayerNorm, the module forms:

$$\boldsymbol{Q} = \boldsymbol{W}_q(\mathbf{LN}(\mathbf{z})), \ \boldsymbol{K} = \boldsymbol{W}_k(\mathbf{LN}(\mathbf{r})), \ \boldsymbol{V} = \boldsymbol{W}_v(\mathbf{LN}(\mathbf{r})), \tag{9}$$

splits into $\alpha$ heads, and computes masked attention weights over the flattened memory positions:

$$\mathbf{Attn} = \texttt{MaskedSoftmax}(\frac{\boldsymbol{Q}\boldsymbol{K}^\intercal}{\sqrt{D/\alpha}}; \mathbf{m}^r). \tag{10}$$

The attention output is $\boldsymbol{Y} = \mathbf{Attn}\boldsymbol{V}$, followed by an output projection $\boldsymbol{W}_{\text{out}}$ and a learned scalar sigmoid gate $c \in \mathbb{R}$:

$$\mathbf{z} = \mathbf{z} + \sigma(c)\boldsymbol{W}_{\text{out}}\boldsymbol{Y}. \tag{11}$$

Our cross-attention conditioning is effective as it injects the retrieved reference as a token level external memory, letting each latent token selectively attend to the most relevant reference positions rather than relying on a coarse global conditioning vector. The sigmoid gated residual makes this guidance stable and adaptive, i.e. starting weak and strengthening only when it improves generation, so the model can leverage retrieval without overwhelming the base dynamics. More details are discussed in Appendix Section B.3.

## 5 Experiments

### 5.1 Experimental Setup

**Datasets.** Following Xue et al. (2023), we run experiments on five standard temporal point process datasets: Amazon (Ni et al., 2019) with 16 event types; Retweet (Zhou et al., 2013) with 3 event types; StackOverflow

(Leskovec & Krevl, 2014) with 22 event types; Taobao (Xue et al., 2022) with 20 event types; Taxi (Whong, 2014) with 10 event types. We also evaluate ReDiTT on two action datasets: Breakfast (Kuehne et al., 2014) with 177 action classes, we scale the timestamp by dividing by 100 to avoid VAE training loss explosion; Multithumos (Yeung et al., 2018) with 65 action classes, we take the 212 sequences with length $\leq 100$ out of total 289 sequences to avoid model ran out of memory. More details are in Appendix Section B.1 and Table 6.

**Baselines.** We compare ReDiTT with four kinds of state-of-the-art TPP models for asynchronous time series: ① RNN-based models, including RMTPP (Du et al., 2016) and NHP (Mei & Eisner, 2017); ② Attention-based models, including SAHP (Zhang et al., 2020), THP (Zuo et al., 2020), AttNHP (Yang et al., 2021), and DTPP (Panos, 2024); ③ Diffusion-based models, including Add&Thin (Lüdke et al., 2023) and ADiff4TPP (Mukherjee et al., 2025); ④ Other popular models including IFTPP (Shchur et al., 2020) and HYPRO (Xue et al., 2022).

**Metrics.** Following Xue et al. (2023), we examine our models on next event prediction. We evaluate the time prediction by root mean square error (RMSE) between predicted time and true time, and the event type prediction by accuracy between predicted type and true type. Note to compute the accumulated RMSE and accuracy, when predicting $\hat{\mathbf{x}}_{i+2}$, the model uses the true history $\{\mathbf{x}_1, \cdots, \mathbf{x}_i, \mathbf{x}_{i+1}\}$ rather than a previously predicted event $\{\mathbf{x}_1, \cdots, \mathbf{x}_i, \hat{\mathbf{x}}_{i+1}\}$. We also examine ReDiTT on long-horizon prediction by computing the optimal transport distance (OTD), a measurement of edit distance between the predicted $\{\hat{\mathbf{x}}_{i+1}, \cdots, \hat{\mathbf{x}}_{i+m}\}$ and the ground truth $\{\mathbf{x}_{i+1}, \cdots, \mathbf{x}_{i+m}\}$. We follow the implementation of OTD by Mei et al. (2019) with dynamic programming to efficiently find alignment and compute distance between predictions and true events. Following Mukherjee et al. (2025), we compute the OTD with horizon window size $m = 5, 10, 20, 30$ respectively.

To facilitate reproducibility, we provide full implementation details and hyperparameter settings in Appendix Section B.

## 5.2   Next-event Prediction and Long Horizon Prediction

In this section, we evaluate our model against existing methods on next-event prediction and long-horizon prediction tasks. Overall, **ReDiTT outperforms existing baselines on both tasks and is particularly effective in regimes with long event sequences and limited training samples**, where purely parametric models struggle to capture rare transitions and long-range temporal dependencies. By retrieving trajectory-level neighbors as explicit context, ReDiTT provides strong guidance for coherent long-horizon generation under sparse supervision. Beyond retrieval itself, the results demonstrate that the manner in which retrieved information is injected into the model is critical. The proposed conditioning strategy, based on cross attention in diffusion transformers, enables the denoising network to selectively attend to the most relevant retrieved context, resulting in more accurate and coherent predictions.

**Next-event prediction.**   Table 1 summarizes the main results on seven benchmark datasets. Performance is evaluated on next-event prediction, using RMSE for time prediction and accuracy for type prediction. Overall, ReDiTT exhibits the strongest and most consistent performance across a broad range of dynamics. It achieves the best RMSE and the highest type accuracy on all datasets. ReDiTT substantially outperforms both classic neural TPP baselines and recent diffusion-based or flow-based forecasters. In addition, ReDiTT delivers large gains in average performance across datasets compared to the strongest baseline, ADiff4TPP (Mukherjee et al., 2025). The average RMSE is reduced by 22%, decreasing from 3.887 for ADiff4TPP to 3.008 for ReDiTT. At the same time, the average type accuracy increases by 12 percentage points, from 42.4% to 55.2%. The advantages of ReDiTT are especially pronounced on Breakfast and MultiThumos, which feature large event vocabularies but substantially fewer training sequences. In this low-data, high-cardinality regime, purely parametric models are prone to sparse supervision for rare event types. Retrieval provides valuable exemplar-based guidance by exposing the model to structurally similar trajectories and richer type co-occurrence patterns. On Breakfast, which contains 177 event types, ReDiTT improves type accuracy from 8.2% to 15.1%. A consistent trend is observed on MultiThumos with 65 event types, where ReDiTT yields a large accuracy gain from 16.3% to 25.0%. These results indicate that retrieval is particularly effective when the label space is large and training coverage per class is limited.

Table 1: **Next-event prediction and long-horizon prediction results** for ReDiTT and existing baselines on seven benchmark datasets. RMSE is computed on the predicted inter-event time, while Accuracy ($\uparrow$) is computed on the predicted event type. Long-horizon forecasting performance is measured by OTD ($\downarrow$) at horizons $m = 5, 10, 20, 30$. We use ADiff4TPP as the unconditional diffusion baseline and include additional baselines provided by Xue et al. (2023). When available, we report the results published in Mukherjee et al. (2025) for direct comparison. The **best results** are highlighted in bold, and the second best results are underlined. The numbers in grey within parentheses indicate the standard deviation with three random seeds 1,2,3.

| | Amazon | | Retweet | | StackOverflow | | Taobao | | Taxi | | Breakfast | | MultiThumos | |
|---|---|---|---|---|---|---|---|---|---|---|---|---|---|---|
| | RMSE ($\downarrow$) | Acc% ($\uparrow$) | RMSE ($\downarrow$) | Acc% ($\uparrow$) | RMSE ($\downarrow$) | Acc% ($\uparrow$) | RMSE ($\downarrow$) | Acc% ($\uparrow$) | RMSE ($\downarrow$) | Acc% ($\uparrow$) | RMSE ($\downarrow$) | Acc% ($\uparrow$) | RMSE ($\downarrow$) | Acc% ($\uparrow$) |
| RMTPP | 0.559 (0.014) | 29.6 (0.008) | 26.207 (5.650) | 51.6 (0.030) | 1.246 (0.293) | 42.4 (0.002) | 0.257 (0.073) | 43.6 (0.000) | 0.351 (0.042) | 88.5 (0.002) | 1.080 (0.301) | 4.8 (0.001) | 7.357 (0.051) | 0.1 (0.000) |
| NHP | 0.640 (0.002) | 30.0 (0.001) | 22.511 (0.033) | 53.9 (0.001) | 1.324 (0.359) | 26.8 (0.146) | 0.168 (0.098) | 49.3 (0.012) | 0.342 (0.075) | 87 (0.007) | 1.168 (0.050) | 4.9 (0.001) | 7.309 (0.005) | 9.9 (0.002) |
| SAHP | 0.517 (0.008) | 32.0 (0.005) | 21.708 (0.001) | 54.0 (0.001) | 1.327 (0.002) | 42.3 (0.002) | 0.154 (0.083) | 46.4 (0.009) | 0.335 (0.175) | 88.1 (0.016) | 1.130 (0.002) | 4.8 (0.000) | 7.674 (0.102) | 10.3 (0.018) |
| THP | 0.550 (0.010) | 34.6 (0.002) | 26.176 (0.050) | 59.5 (0.001) | 1.424 (0.012) | 42.0 (0.013) | 0.314 (0.044) | 45.6 (0.017) | 0.375 (0.065) | 87.3 (0.011) | 1.134 (0.052) | 1.3 (0.000) | 9.254 (0.625) | 13.0 (0.016) |
| AttNHP | 0.755 (0.185) | 30.9 (0.011) | 22.296 (1.135) | 57.2 (0.041) | 1.350 (0.018) | 44.6 (0.001) | 0.280 (0.065) | 47.1 (0.019) | 0.429 (0.012) | 85.2 (0.027) | 1.123 (0.027) | 4.7 (0.001) | 7.887 (0.091) | 14.9 (0.013) |
| IFTPP | 0.465 (0.001) | 35.1 (0.001) | 22.198 (0.234) | 60.0 (0.002) | 1.884 (0.043) | 45.5 (0.008) | 0.598 (0.103) | 55.9 (0.003) | 0.357 (0.013) | 91.4 (0.004) | 3.654 (0.134) | 9.2 (0.003) | 8.980 (0.092) | 14.3 (0.014) |
| DTPP | 0.619 (0.104) | 34.5 (0.002) | 24.680 (6.364) | 59.7 (0.003) | 1.780 (0.167) | 39.3 (0.002) | 0.587 (0.031) | 46.7 (0.003) | 0.302 (0.043) | 87.9 (0.012) | 1.231 (0.089) | 3.5 (0.002) | 8.762 (0.092) | 13.8 (0.008) |
| Add&Thin | 0.461 (0.017) | - | 22.914 (0.348) | - | 1.469 (0.238) | - | 0.440 (0.035) | - | 0.368 (0.015) | - | 1.351 (0.091) | - | 6.081 (0.076) | - |
| HYPRO | 0.583 (0.012) | 33.8 (0.004) | 20.562 (1.633) | 60.0 (0.049) | 1.417 (0.253) | 44.9 (0.002) | 0.307 (0.029) | 55.1 (0.004) | 0.383 (0.008) | 86.5 (0.018) | 1.153 (0.064) | 9.5 (0.003) | 6.481 (0.009) | 15.9 (0.002) |
| ADiff4TPP | 0.413 (0.005) | 33.7 (0.003) | 17.480 (0.023) | 60.7 (0.001) | 1.524 (0.003) | 34.8 (0.013) | 0.140 (0.054) | 57.4 (0.011) | 0.309 (0.003) | 85.6 (0.029) | 1.360 (0.006) | 8.2 (0.001) | 5.981 (0.003) | 16.3 (0.000) |
| ReDiTT (k=1) | 0.410 (0.004) | 49.5 (0.004) | 15.238 (0.081) | 69.6 (0.012) | 1.240 (0.001) | 41.9 (0.005) | **0.134** (0.002) | 58.8 (0.008) | 0.253 (0.002) | 92.8 (0.009) | 1.210 (0.003) | 10.8 (0.002) | 5.800 (0.002) | 17.2 (0.000) |
| **ReDiTT (k=7)** | **0.352** (0.002) | **60.9** (0.000) | **13.429** (0.020) | **78.5** (0.005) | **1.048** (0.001) | **53.0** (0.002) | **0.140** (0.000) | **59.3** (0.001) | **0.239** (0.002) | **94.5** (0.001) | **1.054** (0.015) | **15.1** (0.000) | **4.797** (0.102) | **25.0** (0.000) |

| | OTD ($\downarrow$) | OTD ($\downarrow$) | OTD ($\downarrow$) | OTD ($\downarrow$) | OTD ($\downarrow$) | OTD ($\downarrow$) | OTD ($\downarrow$) |
|---|---|---|---|---|---|---|---|
| RMTPP | 9.9 / 19.7 / 39.2 / 58.4 0.016 / 0.035 / 0.075 / 0.120 | 10.0 / 21.0 / 40.0 / 59.8 0.003 / 0.004 / 0.012 / 0.021 | 9.5 / 18.7 / 37.0 / 54.8 0.075 / 0.160 / 0.320 / 0.487 | 8.7 / 15.9 / 28.7 / 40.6 0.097 / 0.196 / 0.506 / 0.746 | 4.0 / 6.0 / 9.2 / 12.2 0.136 / 0.196 / 0.326 / 0.449 | 9.8 / 19.2 / 34.2 / 45.5 0.076 / 0.204 / 0.712 / 0.808 | 10.0 / 20.0 / 36.8 / 49.0 0.002 / 0.002 / 0.502 / 0.614 |
| NHP | 9.9 / 19.7 / 39.2 / 58.4 0.015 / 0.035 / 0.075 / 0.119 | 10.0 / 20.0 / 39.7 / 59.3 0.003 / 0.008 / 0.028 / 0.056 | 9.5 / 18.8 / 37.2 / 55.2 0.073 / 0.156 / 0.306 / 0.462 | 8.5 / 15.2 / 28.1 / 39.3 0.099 / 0.248 / 0.523 / 0.766 | 4.0 / 5.9 / 9.2 / 12.1 0.137 / 0.199 / 0.327 / 0.452 | 9.8 / 10.2 / 34.0 / 45.2 0.103 / 0.210 / 0.692 / 0.845 | 9.9 / 19.7 / 35.9 / 47.7 0.095 / 0.198 / 0.492 / 0.781 |
| SAHP | 9.9 / 19.7 / 39.1 / 58.3 0.016 / 0.035 / 0.080 / 0.124 | 10.0 / 20.0 / 39.9 / 59.6 0.002 / 0.006 / 0.020 / 0.038 | 9.5 / 18.8 / 37.2 / 55.2 0.072 / 0.156 / 0.304 / 0.459 | 8.6 / 15.5 / 28.3 / 39.9 0.101 / 0.247 / 0.523 / 0.735 | 5.5 / 7.6 / 12.8 / 17.5 0.113 / 0.162 / 0.231 / 0.413 | 10.0 / 19.5 / 34.7 / 46.1 0.002 / 0.196 / 0.625 / 0.817 | 10.0 / 20.0 / 36.7 / 48.8 0.002 / 0.001 / 0.712 / 0.975 |
| THP | 9.9 / 19.7 / 39.1 / 58.2 0.017 / 0.037 / 0.082 / 0.130 | 10.0 / 20.0 / 39.7 / 59.2 0.003 / 010 / 0.032 / 0.063 | 9.5 / 18.8 / 37.1 / 54.9 0.069 / 0.148 / 0.306 / 0.469 | 8.7 / 15.8 / 28.8 / 40.7 0.095 / 0.234 / 0.499 / 0.735 | 6.0 / 10.8 / 14.3 / 17.6 0.094 / 0.163 / 0.318 / 0.388 | 10.0 / 19.6 / 34.9 / 46.3 0.001 / 0.419 / 0.518 / 0.800 | 9.7 / 19.4 / 35.6 / 46.9 0.132 / 0.593 / 0.613 / 0.913 |
| AttNHP | 9.9 / 19.7 / 39.1 / 58.2 0.018 / 0.039 / 0.083 / 0.130 | 10.0 / 20.0 / 39.7 / 59.2 0.005 / 0.012 / 0.032 / 0.065 | 9.4 / 18.7 / 36.9 / 54.4 0.073 / 0.157 / 0.323 / 0.492 | 8.2 / 14.7 / 24.0 / 38.4 0.105 / 0.247 / 0.499 / 0.723 | 4.0 / 5.8 / 9.0 / 11.7 0.105 / 0.247 / 0.240 / 0.307 | 9.8 / 19.2 / 34.0 / 45.1 0.102 / 0.419 / 0.592 / 0.710 | 9.9 / 19.9 / 35.7 / 47.2 0.094 / 0.258 / 0.540 / 0.714 |
| IFTPP | 9.9 / 19.7 / 40.0 / 58.1 0.018 / 0.042 / 0.091 / 0.140 | 10.0 / 20.0 / 39.7 / 59.2 0.003 / 0.011 / 0.031 / 0.064 | 9.4 / 18.6 / 36.8 / 54.3 0.077 / 0.164 / 0.331 / 0.500 | 8.0 / 14.2 / 26.2 / 37.6 0.104 / 0.245 / 0.503 / 0.703 | 4.5 / 5.6 / 8.6 / 11.6 0.098 / 0.162 / 0.221 / 0.337 | 8.8 / 16.7 / 28.5 / 37.7 0.009 / 0.219 / 0.493 / 0.712 | 9.9 / 19.6 / 35.0 / 46.4 0.104 / 0.278 / 0.515 / 0.740 |
| DTPP | 6.9 / 13.7 / 27.6 / 41.7 0.010 / 0.024 / 0.064 / 0.161 | 10.0 / 19.4 / 29.8 / 39.6 0.004 / 0.009 / 0.003 / 0.063 | 7.6 / 15.0 / 29.2 / 43.4 0.020 / 0.036 / 0.036 / 0.486 | 6.8 / 15.1 / 32.3 / 50.0 0.019 / 0.032 / 0.136 / 0.684 | 3.0 / 6.8 / 13.8 / 20.7 0.013 / 0.019 / 0.145 / 0.482 | 8.9 / 17.2 / 26.9 / 39.0 0.010 / 0.301 / 0.391 / 0.794 | 10.0 / 19.3 / 28.9 / 30.1 0.113 / 0.173 / 0.713 / 0.590 |
| HYPRO | 7.0 / 13.0 / 26.1 / 34.1 0.016 / 0.055 / 0.140 / 0.155 | 9.0 / 19.7 / 30.7 / 39.0 0.003 / 0.005 / 0.063 / 0.037 | 7.3 / 12.2 / 29.2 / 36.8 0.017 / 0.028 / 0.454 / 0.563 | 5.8 / 11.4 / 20.8 / 30.7 0.028 / 0.047 / 0.074 / 0.198 | 3.4 / 5.8 / 10.0 / 13.9 0.023 / 0.034 / 0.056 / 0.072 | 8.7 / 17.4 / 27.9 / 32.5 0.008 / 0.029 / 0.294 / 0.009 | 9.4 / 17.9 / 24.4 / 27.9 0.009 / 0.319 / 0.028 / 0.376 |
| ADiff4TPP | 6.2 / 12.4 / 24.7 / 32.9 0.049 / 0.115 / 0.141 / 0.477 | 9.1 / 17.7 / 28.0 / 31.7 0.017 / 0.051 / 0.170 / 0.208 | 6.5 / 12.0 / 22.3 / 30.1 0.400 / 0.922 / 2.142 / 1.789 | 4.9 / 9.9 / 20.4 / 31.1 0.086 / 0.065 / 0.149 / 0.131 | 2.4 / 4.0 / 6.8 / 9.4 0.022 / 0.015 / 0.049 / 0.012 | 8.8 / 17.1 / 24.3 / 28.9 0.090 / 0.102 / 0.139 / 0.158 | 8.2 / 13.6 / 21.1 / 20.2 0.092 / 0.132 / 0.156 / 0.198 |
| ReDiTT (k=1) | 5.9 / 11.5 / 21.3 / 27.8 0.032 / 0.108 / 0.110 / 0.329 | 9.1 / 17.3 / 26.9 / 30.1 0.016 / 0.039 / 0.159 / 0.197 | 6.1 / 10.8 / 19.4 / 27.4 0.389 / 0.893 / 1.210 / 1.623 | 4.8 / 9.8 / 19.8 / 30.7 0.020 / 0.080 / 0.128 / 0.129 | 2.6 / 4.4 / 7.9 / 12.7 0.025 / 0.192 / 0.419 / 0.912 | 8.7 / 16.1 / 22.1 / 27.0 0.095 / 0.200 / 0.138 / 0.200 | 8.2 / 14.3 / 20.6 / 19.6 0.002 / 0.042 / 0.167 / 0.175 |
| **ReDiTT (k=7)** | **5.7** / **10.7** / **19.7** / **25.8** 0.041 / 0.100 / 0.140 / 0.301 | **9.1** / **17.2** / **26.5** / **29.6** 0.018 / 0.042 / 0.112 / 0.124 | **6.0** / **10.7** / **18.9** / **27.4** 0.291 / 0.420 / 0.910 / 1.394 | **4.6** / **9.4** / **19.4** / **29.5** 0.031 / 0.086 / 0.159 / 0.171 | **3.5** / **5.0** / **9.2** / **17.6** 0.005 / 0.209 / 0.468 / 0.923 | **8.2** / **15.4** / **21.6** / **26.4** 0.005 / 0.100 / 0.142 / 0.179 | **8.2** / **14.1** / **20.6** / **19.6** 0.002 / 0.032 / 0.095 / 0.128 |

**Long horizon prediction.** For long-horizon prediction, ReDiTT consistently reduces OTD at horizons $m = 5, 10, 20, 30$ on six datasets compared to the strongest diffusion baseline (Mukherjee et al., 2025). This result demonstrates that retrieval guidance substantially improves sequence-level coherence beyond single-step prediction. The gains are most pronounced on Amazon, where ReDiTT lowers OTD from 12.4 to 10.7 at horizon 10 and from 32.9 to 25.8 at horizon 30, indicating that the advantages of retrieval-augmented conditioning grow as the prediction horizon increases. Similar patterns are observed on Retweet and StackOverflow, where ReDiTT consistently achieves the best OTD across horizons, highlighting the robustness of the approach under diverse temporal dynamics. These long-horizon improvements are driven by the retrieval-conditional design, which injects trajectory-relevant training examples as explicit context. This design enables the model to generate multiple future events coherently while preserving temporal ordering and capturing long-range sequential structure. On Taxi, ReDiTT achieves the best RMSE and accuracy, while OTD is optimized at smaller retrieval sizes. This behavior aligns with the dataset characteristics, which include short sequences and a limited event vocabulary. In this setting, retrieval provides smaller incremental benefits because sequence-level alignment is already well constrained. Overall, these findings support the core motivation of retrieval-augmented conditioning. By injecting trajectory-relevant training examples as explicit context, the model is more effectively guided toward plausible future dynamics.

Table 2: Analysis of different conditioning architecture of DiT for five datasets with a fixed $k = 1$. For unconditional training, we refer to ADiff4TPP.

| | condition | RMSE (↓) | Acc% (↑) | OTD (↓) |
|---|---|---|---|---|
| | uncondition | 0.413 | 33.7 | 6.2 / 12.4 / 24.7 / 32.9 |
| Amazon | adaLN | 0.460 | **51.0** | 7.7 / 15.2 / 28.7 / 36.1 |
| | crossattn | **0.410** | 49.5 | **5.9 / 11.5 / 21.3 / 27.8** |
| | uncondition | 17.480 | 60.7 | 9.1 / 17.7 / 28.0 / 31.7 |
| Retweet | adaLN | 19.266 | 69.4 | 9.6 / 18.9 / 30.1 / 33.9 |
| | crossattn | **15.238** | **69.6** | **9.1 / 17.3 / 26.9 / 30.1** |
| | uncondition | 1.524 | 34.8 | 6.5 / 12.0 / 22.3 / 30.1 |
| StackOverflow | adaLN | **1.012** | **51.3** | 9.4 / 18.8 / 37.3 / 55.5 |
| | crossattn | 1.240 | 41.9 | **6.1 / 10.8 / 19.4 / 27.4** |
| | uncondition | 0.140 | 57.4 | 4.9 / 9.9 / 20.4 / 31.1 |
| Taobao | adaLN | 0.268 | **60.5** | 5.6 / 10.9 / 22.1 / 32.2 |
| | crossattn | **0.134** | 58.8 | **4.8 / 9.8 / 19.8 / 30.7** |
| | uncondition | 0.309 | 85.6 | **2.4 / 4.0 / 6.8 / 9.4** |
| Taxi | adaLN | 0.263 | **94.0** | 6.8 / 11.8 / 23.4 / 34.7 |
| | crossattn | **0.253** | 92.8 | 2.6 / 4.4 / 7.9 / 12.7 |

## 5.3 Ablation Studies

**Different Conditioning Architecture.** As discussed in Peebles & Xie (2023), adaptive layer norm often outperforms cross-attention conditioning in diffusion transformers for class-labeled image generation. We therefore conduct an ablation on five datasets with fixed $k = 1$ to analyze which conditioning architecture better suits asynchronous event sequences. Following the design in Peebles & Xie (2023), we implement adaLN conditioning on $\mathbf{r}$ and also consider Concat$(s, \mathbf{r})$, which concatenates the diffusion timestamp $s$ with $\mathbf{r}$ to encode temporal information, i.e. three paths for $s$, $\mathbf{r}$ and the fused Concat$(s, \mathbf{r})$, each is controlled by a scale hyperparameter. More details can be found in Appendix Section B.3. As reported in Table 2, cross-attention consistently yields stronger long-horizon behavior, achieving substantially lower OTD, whereas adaLN only improves event-type accuracy on three of the five datasets and often degrades sequence-level alignment. StackOverflow illustrates the trade-off clearly. adaLN achieves better next-event metrics, but its long-horizon alignment is much worse. In contrast, cross-attention produces more coherent multi-step forecasts (OTD 10.8 at horizon 10 and 27.4 at horizon 30), even though its next event prediction accuracy is lower and RMSE is higher. For the other three datasets except for Taxi, we observe a similar pattern: any gains adaLN provides in next-event RMSE or type accuracy are relatively modest, while its long-horizon OTD deteriorates substantially. This indicates that adaLN's improvements are largely limited to short-term, local prediction, whereas cross-attention is far more reliable for preserving sequence-level alignment over extended horizons.

We attribute this difference to the mismatch between the inductive bias of adaLN and the structure of conditioning signals in asynchronous time series. In class-labeled image generation, the condition is typically a single global discrete cue shared across the entire sample; in this setting, adaLN's global, layer-wise modulation can effectively inject the label signal and amplify class conditional priors. In contrast, for asynchronous time series, retrieved input-like references are high-entropy and event-dependent, with information that varies across events and time. Cross-attention therefore provides a more suitable mechanism by enabling state-dependent access to context and selectively integrating temporal patterns as the generated history evolves. Overall, the observed trade-off suggests that adaLN primarily strengthens categorical discrimination, whereas cross-attention better supports precise, context-aligned modeling of event times. Moreover, adaLN is also more expensive in our implementation, resulting in a 294M parameter model, whereas the cross-attention variant is substantially lighter with only 209M parameters. Cross-attention also converges faster in practice, reaching strong performance in fewer training iterations. To obtain both predictive quality and computational efficiency, we adopt cross-attention conditioning in our final model.

**Different Aggregation of Retrieved References.** We further compare two strategies for incorporating the top $k$ retrieved references during conditioning: average pooling and concatenation. We perform the experiments on StackOverflow, Breakfast, and MultiThumos, and the results are illustrated in Table 3. Under a fixed retrieval budget $k$, we observe that average pooling of the retrieved references consistently

Table 3: Analysis of the different aggregation strategy and different $k$ for retrieved references on Stack-Overflow, Breakfast, and MultiThumos. Overall, increasing $k$ leads to improved performance, and average pooling outperforms concatenation as $k$ increases.

| | | StackOverflow | | | Breakfast | | | MultiThumos | | |
| --- | --- | --- | --- | --- | --- | --- | --- | --- | --- | --- |
| | | RMSE (↓) | Acc% (↑) | OTD (↓) | RMSE (↓) | Acc% (↑) | OTD (↓) | RMSE (↓) | Acc% (↑) | OTD (↓) |
| uncondition | | 1.524 | 34.8 | 6.5 / 12.0 / 22.3 / 30.1 | 1.360 | 8.2 | 8.8 / 17.1 / 24.3 / 28.9 | 5.981 | 16.3 | 8.2 / 13.6 / 21.1 / 20.2 |
| $k = 1$ | | 1.240 | 41.9 | 6.1 / 10.8 / 19.4 / 27.4 | 1.210 | 10.8 | 8.7 / 16.1 / 22.1 / 27.0 | 5.800 | 17.2 | 8.2 / 14.3 / 20.6 / 19.6 |
| concatenation | $k = 3$ | 1.244 | 42.1 | 6.2 / 10.9 / 19.6 / 27.8 | 1.133 | 8.6 | 8.8 / 16.4 / 22.6 / 27.1 | 7.020 | 15.6 | 8.6 / 14.4 / 20.6 / 20.3 |
| | $k = 5$ | 1.174 | 47.1 | 6.1 / 10.9 / 19.7 / 27.9 | 1.246 | 9.3 | 8.9 / 16.2 / 21.6 / 26.1 | 6.680 | 16.2 | 8.4 / 14.3 / 20.5 / 19.8 |
| | $k = 7$ | 1.163 | 44.9 | 6.0 / 11.1 / 19.7 / 27.6 | 1.211 | 8.3 | 8.6 / 16.2 / 22.5 / 26.3 | 6.083 | 17.3 | 8.4 / 14.1 / 19.6 / 19.7 |
| average pooling | $k = 3$ | 1.126 | 48.7 | 6.0 / 10.7 / 19.2 / 27.8 | 1.197 | 12.5 | 8.7 / 15.8 / 21.7 / 27.5 | 6.312 | 18.3 | 8.3 / 14.4 / 21.0 / 20.3 |
| | $k = 5$ | 1.101 | 50.6 | 6.0 / 10.7 / 19.4 / 28.8 | 1.113 | **15.3** | **8.4 / 15.4 / 21.3 / 26.4** | 4.956 | 22.0 | 8.5 / 14.3 / 20.7 / 20.0 |
| | $k = 7$ | **1.048** | **53.0** | **6.0 / 10.7 / 18.9 / 27.4** | **1.054** | 15.1 | 8.2 / 15.4 / 21.6 / 26.4 | **4.797** | **25.0** | **8.2 / 14.1 / 20.6 / 19.6** |

outperforms concatenation across three metrics. Moreover, as $k$ increases, the performance gap widens, mostly notably on event-type prediction accuracy. For concatenation, the accuracy at $k = 7$ is lower than at $k = 5$ for StackOverflow and Breakfast. We hypothesize that pooling becomes increasingly beneficial with larger retrieved sets because it acts as a robust aggregation operator, that being said, it emphasizes signals that are consistent across references with high similarity weights while dampening irrelevant information. In contrast, concatenation scales the conditioning length linearly with $k$, which increases exposure to irrelevant retrievals and makes it easier for the model to overfit to noisy reference. The results showed that this effect becomes more obvious as $k$ grows, leading to a larger degradation in categorical prediction. We therefore adopt average pooling in our conditioning implementation.

**Different $k$ for Retrieval Mechanism.** We also conducted an ablation study on the number of retrieved references $k$ to further investigate how the amount of retrieved context influences conditioning effectiveness while keeping all other settings fixed, specifically, whether a small set of references provides sufficient guidance for generation, or whether increasing $k$ introduces irrelevant matches that act as noise and degrade performance. As listed in Table 3, larger $k$ does not improves the results for concatenation conditioning much. However, for average pooling conditioning, increasing $k$ consistently improves both RMSE and event type accuracy, whereas OTD remains largely stable across different values of $k$. This suggests that retrieving more neighbors can provide additional contextual signal that helps the model refine local predictions, particularly for event type inference, while preserving the global sequence level structure captured by OTD. Notably, for StackOverflow, event type accuracy reaches 53% at $k = 7$, representing a 10.9 point improvement over $k = 1$. This indicates that multi-reference conditioning is beneficial, and a moderate retrieval size can yield significant gains, especially for categorical prediction, while preserving long-horizon sequence alignment.

**Effect of Time and Type Conditioning Components.** We further ablate the conditioning signal by using time-only or event-type-only guidance. Because retrieval is performed in the latent space, we first retrieve reference sequences via latent similarity, and then mask out the undesired modality by replacing the corresponding latent block (time or type) with a zero latent (i.e., the latent obtained by encoding an all-zero input in the original space), while keeping the other block unchanged. This design isolates the contribution of each modality without altering the retrieval set.

We evaluate this ablation on Taobao (20 event types), Breakfast (177 event types), and Multithumos (64 event types) with fixed $k = 1$. As shown in Table 4, each modality contributes useful but incomplete information. Time-only conditioning can improve metrics that are more sensitive to temporal dynamics, while event-type-only conditioning can perform better on metrics that depend more on categorical structure. However, neither single component consistently dominates across all evaluation criteria, indicating that time and type provide complementary guidance. When both conditioning blocks are used together, the model achieves strong and balanced performance across all metrics, suggesting that jointly conditioning on temporal and event-type information enables the retrieval guidance to capture both when events occur and what events occur. This supports the design choice of using the full time-and-type conditioning signal rather than relying on either modality alone.

Table 4: Analysis of different conditioning components with a fixed $k = 1$ on Taobao, Breakfast, and MultiThumos. Overall, conditioning on both event time and event type outperforms conditioning on either time or type alone.

| | Taobao | | | Breakfast | | | MultiThumos | | |
|---|---|---|---|---|---|---|---|---|---|
| | RMSE (↓) | Acc% (↑) | OTD (↓) | RMSE (↓) | Acc% (↑) | OTD (↓) | RMSE (↓) | Acc% (↑) | OTD (↓) |
| time & type | **0.134** | 58.5 | **4.8 / 9.8 / 19.8 / 30.7** | **1.210** | 10.8 | 8.7 / 16.1 / 22.1 / 27.0 | **5.800** | 17.2 | **8.2 / 14.3 / 20.6 / 19.6** |
| time only | 0.135 | 59.1 | 4.7 / 9.9 / 20.3 / 31.2 | 1.312 | 5.5 | **8.3 / 15.7 / 19.6 / 25.7** | 10.426 | 14.5 | 8.3 / 14.5 / 20.8 / 19.7 |
| type only | 0.134 | **61.1** | 4.9 / 10.2 / 20.9 / 31.5 | 1.575 | 10.3 | 8.6 / 16.7 / 22.2 / 27.6 | 6.101 | **17.9** | 8.2 / 14.3 / 20.7 / 20.1 |

## 6 Conclusion

We propose ReDiTT, a retrieval augmented conditional Diffusion Transformer for asynchronous event sequence forecasting that models the joint uncertainty of the next inter event time and event type. ReDiTT operates in latent space and uses a memory bank to retrieve structurally similar latent sequences as reference conditions during training and inference. The retrieved references are injected through cross attention in DiT blocks, enabling the model to selectively use trajectory specific temporal patterns that are hard to capture with purely parametric conditioning. Across seven real world datasets, ReDiTT achieves state-of-the-art results on both next event prediction and long horizon forecasting. Ablations confirm the value of retrieval based conditioning and cross attention integration, and show that structured guidance is critical for stable long horizon generation in temporal point processes.

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

## A    Discussion

**Limitations.** Although ReDiTT achieves substantial empirical gains, it introduces additional system overhead by requiring a reference bank and performing retrieval at inference time, which increases both memory usage and latency. Another limitation is that gains can be limited on small-scale datasets, especially when the sequence length is relatively short, the model might have limited improvement on capturing long range behaviors.

**Future Work.** Future directions include learning retrieval representations that are better aligned with forecasting objectives, and making conditioning more robust via reference weighting or retrieval dropout. On the efficiency side, we can reduce cost with compressed or hierarchical reference banks and caching. It is also promising to incorporate richer context such as covariates or text, and to explore training objectives that more directly target long-horizon metrics.

**Broader Impact Statement.** This paper presents a diffusion-based model for predicting future events from historical data. Such models may support improved planning and decision-making by enabling probabilistic forecasts that capture uncertainty and multiple plausible futures. This can be beneficial in domains where anticipating diverse outcomes is important. However, predictions of future events can be misinterpreted or misused, particularly if treated as deterministic or deployed in high-stakes settings without appropriate oversight. The model may also inherit biases present in training data and does not address issues of fairness or causality. Additionally, diffusion-based methods can be computationally demanding, and their environmental and efficiency costs should be considered. This work is intended as a methodological contribution, and we encourage responsible use and further research on interpretability, fairness, and efficiency.

## B    Implementation Details

### B.1    Dataset Preparation and Baselines

Following (Xue et al., 2023), we pad Amazon, Retweet, StackOverflow, Taobao, and Taxi to the maximum sequence length within each dataset. For Breakfast, we rescale time values by dividing by 100 to stabilize VAE training. For the two text-based action datasets, Breakfast and MultiThumos, we encode event types as integer indices in the same way as the other datasets, and split the data into train/validation/test sets with a 70/10/20 ratio. Additional dataset details are provided in dataset settings of Table 6.

For baseline models, we use the implementations from EasyTPP (Xue et al., 2023) for RMTPP Du et al. (2016), NHP Mei & Eisner (2017), SAHP Zhang et al. (2020), THP Zuo et al. (2020), AttNHP Yang et al. (2021), and IFTPP Shchur et al. (2020).

For other baselines, we adapt DTPP Panos (2024) to predict inter-event times in their original scale rather than in log space, following ADiff4TPP Mukherjee et al. (2025). For Add&Thin Lüdke et al. (2023), the method outputs only inter-event times, so we do not report event-type accuracy or OTD.

### B.2    VAE Training

Our VAE implementation for asynchronous time series follows ADiff4TPP Mukherjee et al. (2025). For each event $\mathbf{x} = (t, e)$, we train a $\beta$-VAE consisting of an encoder $E_\phi$ that maps $\mathbf{x}$ to a latent variable $\mathbf{z}$, modeled as

a Gaussian distribution parameterized by an inferred mean (and variance). A decoder $D_\phi$ then reconstructs the event from $\mathbf{z}$, producing $\tilde{\mathbf{x}} = (\tilde{t}, \tilde{e}) = D_\phi(\mathbf{z})$. The whole $\beta$-VAE is optimized with the following objective:

$$\mathcal{L} = (t - \tilde{t})^2 + \text{CE}(e, \tilde{e}) + \beta\mathcal{L}_{KL}, \tag{12}$$

$\beta$ controls the weight of the KL regularization term, which penalizes deviations of the latent distribution $\mathcal{X}$ from a standard Gaussian prior. As shown in the VAE settings of Table 6, we use the same $\beta$ choices reported in the appendix of Mukherjee et al. (2025).

### B.3 DiT Implementation

**Adapting DiT for Asynchronous Time Series.** Following Peebles & Xie (2023), we instantiate $v_\theta(\cdot, \boldsymbol{A}(s))$ with a Diffusion Transformer (DiT) backbone, since its Transformer blocks provide a flexible way to model long-range dependencies while remaining compatible with diffusion or flow-style training objectives. We only make minimal modifications to adapt DiT from images to asynchronous event sequences. In particular, we remove the image-specific patchify and patch-embedding modules and instead feed the model with latent event tokens produced by the encoder $E_\phi$. This replacement preserves the core DiT computation (stacked Transformer blocks with conditioning) while ensuring that the model operates directly on a sequence of event-level representations, which is the natural input format for asynchronous time series forecasting.

When initializing the model, we specify a hyperparameter $N$ that sets the maximum sequence length the model can represent and generate. Concretely, sequences shorter than $N$ are padded and masked, allowing us to use a fixed token length for efficient batched training and inference while still supporting variable-length histories and horizons. Finally, to inject the conditioning matrix $\boldsymbol{A}(s) \in \mathbb{R}^{N \times N}$, we extend the original DiT conditioning design by constructing an embedding that reflects both temporal progression and structural interactions encoded by $\boldsymbol{A}(s)$. We achieve this by broadcasting the entries of $\boldsymbol{A}(s)$ through sinusoidal frequency features, producing a dense embedding that provides the Transformer blocks with a smooth, scale-aware representation of the matrix. This design allows the network to exploit the temporal and structural dynamics captured by $\boldsymbol{A}(s)$ without introducing additional architectural complexity.

**Timestep Embedding.** We set the maximum period to $T_{\max} = 10000$ and use an embedding horizon of $f = 128$ sinusodial frequencies. Given the diagonal elements of $\boldsymbol{A}(s) \in \mathbb{R}^{N \times N}$, we construct a frequency-scaled argument matrix $\boldsymbol{B} \in \mathbb{R}^{N \times h}$ to encode each event position with multiple time scales. Concretely, for $i \in \{1, \cdots, N\}$ and $j \in \{1, \cdots, h\}$, we define:

$$\boldsymbol{B}_{ij} = \boldsymbol{A}(s)_{ii} \cdot T_{\max}^{-\frac{j-1}{f}} \tag{13}$$

so that $j$ indexes a geometric progression of frequencies spanning from coarse to fine resolutions. We then form the final timestep embedding $\texttt{emb}_s \in \mathbb{R}^{N \times 2f}$ by concatenating sine and cosine features:

$$\texttt{emb}_s = [\cos(\boldsymbol{B}), \sin(\boldsymbol{B})]. \tag{14}$$

Then we map this to the model hidden size $D$ via an MLP:

$$\texttt{emb}_s = \boldsymbol{W}_2 \textbf{SiLU}(\boldsymbol{W}_1 \texttt{emb}_s + b_1) + b_2 \in \mathbb{R}^{N \times D} \tag{15}$$

Following the choice of Mukherjee et al. (2025), we define the schedule function $\boldsymbol{A}$ as:

$$\boldsymbol{A}(s)_{ij} = \begin{cases} 0 & \text{if } i \neq j, \\ \max(0, \min(\frac{2N-i-s(2N-1)}{N}, 1)) & \text{if } i = j, \end{cases} \tag{16}$$

which is a diagonal matrix that assigns a separate noise schedule to each event position. The schedule is constructed so that tokens corresponding to later events receive noise at earlier diffusion times than tokens

for earlier events. Equivalently, later events are corrupted sooner, and the model is trained to denoise them earlier during the reverse process, forcing it to learn forecasting of the sequence tail under higher noise levels.

This sinusoidal construction provides a smooth multi-scale representation of the time conditioning signal $\boldsymbol{A}(s)$, allowing the Transformer blocks to access both low frequency (global) and high frequency (local) temporal variations through a fixed dimension embedding.

**Positional Embedding.** Before entering the DiT blocks, both current latent input and retrieved reference are expanded to hidden size $D$ by channel repetition, for $\mathbf{r}$, we also broadcast the mask $\mathbf{m}^z$ across the channel dimension. Then $\mathbf{z}$ and $\mathbf{r}$ are augmented with fixed positional embeddings $\mathtt{pos}$ to get:

$$\mathbf{z} = \mathrm{Repeat}(\mathbf{z}) + \mathtt{pos_z} \in \mathbb{R}^{N \times D}, \ \mathbf{r} = \mathrm{Repeat}(\mathbf{r}) + \mathtt{pos_r} \in \mathbb{R}^{N \times D}, \tag{17}$$

where $\mathtt{pos}$ is defined as follows. For input sequence $\mathbf{z}$ with $N$ being $\mathtt{num\_rows}$, we assign each position an integer "row index" $p \in \{0, 1, \cdots, N-1\}$. For embedding dimension $D$, define frequency coefficients:

$$\gamma_i = 10000^{-\frac{i}{D/2}}, i = 0, \cdots, \frac{D}{2} - 1. \tag{18}$$

The fixed positional embedding for row $p$ is:

$$\mathtt{pos}(p) = [\sin(p\gamma), \cos(p\gamma)] \in \mathbb{R}^D. \tag{19}$$

Stacking all rows gives:

$$\mathtt{pos} \in \mathbb{R}^{N \times D} \tag{20}$$

Since our retrieved reference $\mathbf{r}$ and input $\mathbf{z}$ has the same dimension, so we apply the same $\mathtt{pos}$ embedding to both of them.

**Mask Mechanism.** When training or sampling sequences whose length $n$ is smaller than the maximum token budget $N$, we apply an attention mask within multi-head self-attention and the multi-head cross attention. so that padded positions do not participate in the computation. The similar strategy is designed in the retrieval process to mask the padded positions. This masking enforces that the Transformer operates only on valid event tokens, preserving the natural ordering of the sequence and preventing information leakage through padding. Combined with our asynchronous noise schedule, the masked attention mechanism aligns the model's computation with the underlying temporal structure of the data. It also makes the framework compatible with variable-length inputs and flexible forecasting horizons, since the same architecture can condition on any observed prefix and generate different prediction window sizes without changing the model.

**AdaLN Conditioning Implementation.** We also include the details of our adaLN conditioning for retrieved asynchronous time series. Inside the DiT block, let $\mathbf{z} \in \mathbb{R}^{N \times D}$ be the token features entering the block, we have three conditioning streams:

- time embedding $\mathtt{emb}_s$,

- reference embedding $\mathbf{r}$,

- fused conditioning $\mathtt{Concat}(s, \mathbf{r})$.

We first apply separate LayerNorms and project the concatenation to the desired dimension:

$$s = \mathbf{LN}_s(s), \ \mathbf{r} = \mathbf{LN_r}(\mathbf{r}), \ \mathrm{cond} = \mathtt{Proj}(\mathtt{Concat}(s, \mathbf{r})) = \boldsymbol{W}_{\mathrm{cond}}[s; \mathbf{r}] + b_{\mathrm{cond}} \in \mathbb{R}^{N \times D}, \tag{21}$$

where $[\,\cdot\,;\,\cdot\,]$ concatenates along channels and $\boldsymbol{W}_{\mathrm{cond}} \in \mathbb{R}^{D \times 2D}$. Each stream produces a modulation vector via an MLP:

$$\boldsymbol{M}_s = f_s(s) \in \mathbb{R}^{N \times 6D}, \ \boldsymbol{M_r} = f_{\mathbf{r}}(\mathbf{r}) \in \mathbb{R}^{N \times 6D}, \ \boldsymbol{M}_{\mathrm{cond}} = f_{\mathrm{cond}}(\mathrm{cond}) \in \mathbb{R}^{N \times 6D}. \tag{22}$$

Then we scale each modulator by a learned scalar passed through a sigmoid:

$$M_s = \sigma(\lambda_s)M_s, \ M_{\mathbf{r}} = \sigma(\lambda_{\mathbf{r}})M_{\mathbf{r}}, \ M_{\text{cond}} = \sigma(\lambda_{\text{cond}})M_{\text{cond}}, \tag{23}$$

with trainable $\lambda_s, \lambda_{\mathbf{r}}, \lambda_{\text{cond}} \in \mathbb{R}$.

The combined modulation is:

$$M = M_s + M_{\mathbf{r}} + a \cdot M_{\text{cond}} \in \mathbb{R}^{N \times 6D}, \text{ where } a \in \{0,1\} \text{ is a flag hyperparameter.} \tag{24}$$

Then $M$ is split into six $D$-dimension tensors and enters the layers as vanilla adaLN in DiT.

**Multi-reference Cross-attention by Concatenating $k$ References into Length $kN$ memory.** The main query is $\mathbf{z} \in \mathbb{R}^{N \times D}$, and in the average pooling case $\mathbf{r} \in \mathbb{R}^{N \times D}$, we just need to adapt to $\mathbf{r} \in \mathbb{R}^{N \times k \times D}$ with per reference masks $\mathbf{m} \in \{0,1\}^{k \times N}$. Similarly to Section 4.3, the only changes are $K \in \mathbb{R}^{N \times k \times D}, V \in \mathbb{R}^{N \times k \times D}$, then split them into $\alpha$ heads with $K \in \mathbb{R}^{\alpha \times (kN) \times \frac{D}{\alpha}}, V \in \mathbb{R}^{\alpha \times (kN) \times \frac{D}{\alpha}}$.

## B.4 Hyperparameters and Computation Cost

We report the dataset specifications, model hyperparameters, and computation costs in Table 6.

The long time reported for next-event prediction mainly comes from the autoregressive evaluation protocol over the full sequence length $N$. Following prior work (Xue et al., 2023; Mukherjee et al., 2025), we compute this metric by iterating $i = 1, \cdots, N-1$ and predicting the next event $\hat{\mathbf{x}}_{i+1}$ conditioned on the prefix $\{\mathbf{x}_1, \cdots, \mathbf{x}_i\}$. Consequently, obtaining the full set of one-step-ahead predictions $\hat{\mathbf{x}}_2, \cdots, \hat{\mathbf{x}}_N$ requires $N-1$ separate forward passes, after which RMSE and accuracy are evaluated between $\hat{\mathbf{x}}$ and $\mathbf{x}$. The detailed algorithm is provided in Appendix E of Mukherjee et al. (2025). We use the implementation from Mukherjee et al. (2025), and this implementation is not optimized for inference; therefore, the reported time could be reduced with further implementation improvements. For a single sequence, ReDiTT takes 2.1s for one-step next-event prediction, compared to 2.0s for the unconditional baseline of Mukherjee et al. (2025). This indicates that the retrieval-conditioned design adds only a reasonable runtime overhead. Because next-event prediction is relatively slow under the standard autoregressive evaluation protocol, this also motivates our long-horizon prediction setting: instead of repeatedly running one-step inference many times, we generate an entire future segment in a single run, enabling more efficient forecasting of long sequences.

We also include the iteration numbers we used in ablation tables in Table 7 and Table 8. Training iteration is indeed one of the main hyperparameters we tuned, and we will clarify the checkpoint-selection protocol in the revision. For each configuration, we trained the model for up to 500k iterations, saved checkpoints every 50k iterations, and evaluated each checkpoint using all three reported criteria: next-event time error, next-event type accuracy, and long-horizon OTD. We then reported the checkpoint that gave strong and balanced performance across these metrics, rather than selecting using only a single metric. Table 5 shows an example for ReDiTT on MultiThumos with $k = 7$. We select the 350k checkpoint because it provides the best overall tradeoff across the three objectives. Although the 300k checkpoint gives the lowest RMSE and later checkpoints slightly improve OTD at some horizons, the 350k checkpoint substantially improves type accuracy while keeping RMSE and OTD competitive.

## B.5 Retrieval overhead.

We further quantify the computational and memory overhead introduced by retrieval. ReDiTT stores a latent reference bank containing the VAE latent tokens of all training sequences, together with their padding masks. For a dataset with $N_{\text{train}}$ training sequences, padded sequence length $N$, latent dimension $d$, and 32-bit floating point storage, the dominant memory cost is

$$N_{\text{train}} \times N \times d \times 4$$

bytes, plus a small mask cost of approximately

$$N_{\text{train}} \times N$$

Table 5: Checkpoint-wise performance of ReDiTT on MultiThumos with $k = 7$. Lower is better for RMSE and OTD; higher is better for accuracy. We choose the 350k checkpoint to have good results for all of the metrics.

| Checkpoint | RMSE ↓ | Accuracy ↑ | OTD@5 ↓ | OTD@10 ↓ | OTD@20 ↓ | OTD@30 ↓ |
|---|---|---|---|---|---|---|
| 50k | 5.4777 | 0.2548 | 8.3745 | 14.4535 | 21.3398 | 21.3327 |
| 100k | 5.0321 | 0.2355 | 8.3314 | 14.6365 | 21.0014 | 19.8137 |
| 150k | 4.8918 | 0.2382 | 8.6275 | 14.2741 | 20.6826 | 20.2415 |
| 200k | 4.8550 | 0.2371 | 8.4456 | 14.2449 | 20.9167 | 20.0028 |
| 250k | 4.8380 | 0.2355 | 8.5000 | 14.3425 | 20.6929 | 19.9026 |
| 300k | **4.7628** | 0.2376 | 8.3457 | 14.1603 | 20.9749 | **19.7040** |
| 350k | 4.7974 | 0.2500 | 8.5046 | 14.2710 | 20.8755 | 19.8915 |
| 400k | 4.8566 | 0.2425 | 8.4571 | **14.0243** | 20.5657 | 19.8726 |
| 450k | 4.8451 | 0.2446 | 8.5062 | 14.0904 | **20.5492** | 19.7714 |

Table 6: ReDiTT Hyperparameters of reported results in Table 1. For the next-event prediction task, inference time is reported using an auto-regressive methodology.

| | | Amazon | Retweet | StackOverflow | Taobao | Taxi | Breakfast | Multithumos |
|---|---|---|---|---|---|---|---|---|
| dataset | # event types | 16 | 3 | 22 | 20 | 10 | 177 | 64 |
| | max length | 94 | 97 | 101 | 64 | 38 | 131 | 100 |
| | # train data | 6454 | 9000 | 1401 | 1300 | 1400 | 179 | 209 |
| | # test data | 1851 | 1520 | 401 | 500 | 400 | 52 | 61 |
| | # validation data | 922 | 1535 | 401 | 200 | 200 | 25 | 27 |
| VAE | latent dimension $d$ | 32 | 32 | 32 | 32 | 32 | 32 | 32 |
| | $\beta_{max}$ | 1e-2 | 1e-1 | 1e-2 | 1e-2 | 1e-2 | 1e-2 | 1e-2 |
| DiT | latent size | 96 | 96 | 96 | 96 | 96 | 96 | 96 |
| | hidden size | 1152 | 1152 | 1152 | 1152 | 1152 | 1152 | 1152 |
| | depth | 7 | 7 | 7 | 7 | 7 | 7 | 7 |
| | # heads | 16 | 16 | 16 | 16 | 16 | 16 | 16 |
| | mlp ratio | 4 | 4 | 4 | 4 | 4 | 4 | 4 |
| train | batch size | 4 | 4 | 4 | 4 | 4 | 4 | 4 |
| | iterations for k=1 | 500k | 550k | 550k | 30k | 950k | 100k | 100k |
| | iterations for k=3 | - | - | 500k | - | - | 300k | 300k |
| | iterations for k=5 | - | - | 500k | - | - | 300k | 300k |
| | iterations for k=7 | 450k | 450k | 550k | 25k | 50k | 400k | 350k |
| | lr | 3e-5 | 3e-5 | 3e-5 | 3e-5 | 3e-5 | 3e-5 | 3e-5 |
| | ema decay | 0.9999 | 0.9999 | 0.9999 | 0.9999 | 0.9999 | 0.9999 | 0.9999 |
| | time per 50k iteration | 50 min | 50 min | 50min | 50min | 50min | 50min | 50min |
| test | total time for next-event prediction | 18h12min | 16h38min | 5h | 1h30min | 14min | 1h20min | 43min |
| | total time for long-horizon prediction | 1h35min | 1h24min | 24min | 11min | 3min | 5min | 4min |
| | GPU | 1 A100 40G | 1 A100 40G | 1 A100 40G | 1 A100 40G | 1 A100 40G | 1 A100 40G | 1 A100 40G |

Table 7: To facilitate reproducibility, we report the training iteration counts associated with the results presented in Table 2.

| | Amazon | Retweet | StackOverflow | Taobao | Taxi |
|---|---|---|---|---|---|
| uncondition | 950k | 950k | 900k | 150k | 900k |
| adaLN | 950k | 950k | 950k | 950k | 950k |
| crossattn | 550k | 550k | 550k | 30k | 950k |

bytes. Thus, the reference-bank memory scales linearly in the number of training sequences, sequence length, and latent dimension.

To measure retrieval latency, we isolate the exact nearest-neighbor retrieval step from the rest of inference. For each dataset, we use a test batch of up to 128 sequences and compute the same masked token-wise cosine similarity used by ReDiTT between the query prefix latents and all training reference latents. The measured operation includes latent-token normalization, the batched similarity computation, masking over valid prefix/reference positions, averaging over valid positions, and the top-$k$ selection with $k = 7$. It does

Table 8: To facilitate reproducibility, we report the training iteration counts associated with the results presented in Table 3 and Table 4.

| | StackOverflow | | Breakfast | | MultiThumos | |
|---|---|---|---|---|---|---|
| | Concat | Avg | Concat | Avg | Concat | Avg |
| $k = 1$ | 550k | 550k | 100k | 100k | 100k | 100k |
| $k = 3$ | 600k | 500k | 300k | 300k | 300k | 300k |
| $k = 5$ | 750k | 500k | 300k | 300k | 300k | 300k |
| $k = 7$ | 550k | 550k | 300k | 400k | 300k | 350k |

| | Taobao | Breakfast | MultiThumos |
|---|---|---|---|
| time & type | 30k | 100k | 100k |
| time only | 50k | 300k | 300k |
| type only | 50k | 300k | 300k |

Table 9: Reference-bank memory and exact-retrieval latency. Memory includes the latent token bank and padding masks. Latency measures exact top-$k$ retrieval for one test batch of up to 128 query sequences, excluding the diffusion solve.

| Dataset | $N_{\text{train}}$ | $N$ | Memory (MB) | Retrieval latency (ms) |
|---|---|---|---|---|
| Amazon | 6454 | 94 | 222.75 | 3.33 |
| Retweet | 9000 | 97 | 320.54 | 5.14 |
| StackOverflow | 1401 | 101 | 51.95 | 0.67 |
| Taobao | 1300 | 64 | 30.55 | 0.36 |
| Taxi | 1400 | 38 | 19.53 | 0.28 |
| Breakfast | 179 | 131 | 8.61 | 0.22 |
| MultiTHUMOS | 209 | 100 | 7.67 | 0.14 |

not include the subsequent diffusion ODE solve, so it measures the overhead of the retrieval step itself. All measurements are performed on a single GPU using the full training reference bank.

Table 9 shows that the reference-bank memory is modest for the benchmark-scale datasets. The largest bank is Retweet, requiring approximately 320.5 MB, followed by Amazon at 222.8 MB. The remaining datasets require substantially less memory, ranging from 7.7 MB to 52.0 MB. Exact retrieval latency is also small relative to the diffusion-based forecasting cost: full-bank top-$k$ retrieval takes about 5.1 ms on Retweet and 3.3 ms on Amazon, and less than 1 ms on the smaller datasets.

The main deployment consideration is scaling to much larger reference banks. Exact retrieval scales linearly with bank size because each query prefix is compared against all stored training sequences:

$$O(B\, N_{\text{train}}\, N\, d),$$

where $B$ is the query batch size. This exact search is practical for the datasets considered here, but for larger deployments the same framework can use approximate nearest-neighbor search, reference-bank subsampling, clustering/prototype selection, or cached retrieval results. Since retrieval is used only to provide conditioning references, these approximations are natural ways to trade a small amount of retrieval precision for lower memory and latency.

## C  Diagnosing the Analog-Forecasting Assumption.

### C.1  When does retrieval help?

Retrieval-augmented forecasting relies on an analog-forecasting assumption: if two observed histories are similar, then their future continuations should also be similar. This assumption is natural for repeated or locally stationary event dynamics, but it may fail when the process is highly stochastic, non-stationary, or when similar prefixes can lead to different futures. We therefore add a diagnostic analysis to evaluate whether the references retrieved by ReDiTT are meaningful future templates.

We use the same retrieval similarity as in the main ReDiTT model. Consider a padded asynchronous sequence $\mathbf{x} = \{\mathbf{x}_1, \ldots, \mathbf{x}_N\}$ with padding mask $\mathbf{m} \in \{0, 1\}^N$, where $N$ is the maximum padded length. The pretrained VAE encoder $E_\phi$ maps the sequence to per-event latent tokens $\mathbf{z} = E_\phi(\mathbf{x}) \in \mathbb{R}^{N \times d}$. We precompute a latent reference bank $\mathcal{B} = \{(\mathbf{z}^r, \mathbf{m}^r) : \mathbf{z}^r = E_\phi(\mathbf{x}^r), \mathbf{x}^r \in \mathcal{X}_{\text{train}}\}$. For a test sequence $\mathbf{x}^q$ with unpadded length $L$,

prediction horizon $m$, and observed prefix length $n = L - m$, we construct a prefix mask $\mathbf{m}^{q,n}$ such that

$$\mathbf{m}_i^{q,n} = \mathbf{1}[i \leq n],$$

with padded positions also masked out. Let $\boldsymbol{q} = E_\phi(\mathbf{x}^q)$ denote the latent tokens of the test sequence. For each candidate reference $(\mathbf{r}, \mathbf{m}^r) \in \mathcal{B}$, we compute the same masked token-wise cosine similarity used in ReDiTT as in Equation (5):

$$\mathtt{sim}(\boldsymbol{q}, \mathbf{r}) = \frac{1}{\sum_{i=1}^{N} \mathbf{1}[\mathbf{m}_i^{q,n} \wedge \mathbf{m}_i^r]} \sum_{i:\mathbf{m}_i^{q,n} \wedge \mathbf{m}_i^r} \langle \hat{\boldsymbol{q}}_i, \hat{\mathbf{r}}_i \rangle,$$

where $\hat{\boldsymbol{q}}_i = \boldsymbol{q}_i / \|\boldsymbol{q}_i\|_2$ and $\hat{\mathbf{r}}_i = \mathbf{r}_i / \|\mathbf{r}_i\|_2$ are $\ell_2$-normalized latent tokens. We then retrieve the top-$k$ references according to this similarity:

$$R(\boldsymbol{q}) = \mathrm{TopK}_{(\mathbf{r}, \mathbf{m}^r) \in \mathcal{B}} \, \mathtt{sim}(\boldsymbol{q}, \mathbf{r}).$$

The only difference from standard ReDiTT inference is that, for this diagnostic, the query mask is explicitly set by the horizon $m$ so that retrieval uses only the observed prefix $\mathbf{x}_{1:n}^q$, while the held-out suffix $\mathbf{x}_{n+1:n+m}^q$ is reserved for evaluating whether the retrieved reference has a similar future.

Let $\mathbf{r}^\star$ be the top-1 retrieved reference according to this similarity:

$$\mathbf{r}^\star = \arg \max_{(\mathbf{r}, \mathbf{m}^r) \in \mathcal{B}} \mathtt{sim}(\boldsymbol{q}, \mathbf{r}).$$

We then evaluate whether the future of the retrieved reference is close to the true future of the query. For test example $j$, with prefix length $n_j = L_j - m$, we compute

$$D_{\mathrm{ret}}^{(j)}(m) = \mathrm{OTD}\left(\mathbf{x}_{n_j+1:n_j+m}^{q,j}, \mathbf{x}_{n_j+1:n_j+m}^{r^\star,j}\right).$$

As a random baseline, we sample a training reference $\mathbf{x}^{r_{\mathrm{rand}},j}$ uniformly at random and compute

$$D_{\mathrm{rand}}^{(j)}(m) = \mathrm{OTD}\left(\mathbf{x}_{n_j+1:n_j+m}^{q,j}, \mathbf{x}_{n_j+1:n_j+m}^{r_{\mathrm{rand}},j}\right).$$

Finally, we report the empirical retrieved/random future distance ratio for each dataset:

$$\rho(m) = \frac{\frac{1}{N_m} \sum_{j=1}^{N_m} D_{\mathrm{ret}}^{(j)}(m)}{\frac{1}{N_m} \sum_{j=1}^{N_m} D_{\mathrm{rand}}^{(j)}(m)},$$

where

$$N_m = |\{j : L_j > m\}|$$

is the number of test sequences with at least $m$ real future events. A value $\rho(m) < 1$ indicates that retrieved futures are closer to the true futures than random training futures. We also report the correlation between the prefix similarity $\mathtt{sim}(\boldsymbol{q}, \mathbf{r}^\star)$ and $D_{\mathrm{ret}}(m)$; a negative correlation indicates that more similar prefixes tend to have closer futures.

**Results.** Table 10 shows that the analog-forecasting assumption holds strongly on some datasets, but not uniformly across all domains. Taobao, StackOverflow, and Breakfast provide the clearest evidence. On Taobao, retrieved futures are much closer than random futures across all horizons, with $\rho(m) \approx 0.73$–0.77, and prefix similarity is strongly negatively correlated with future OTD. StackOverflow shows a similar pattern, with $\rho(m)$ reaching approximately 0.75 at longer horizons, indicating that retrieved references provide useful future templates. Breakfast also supports the assumption, with retrieved futures improving over random futures by roughly 6%–17% depending on the horizon.

Amazon and Retweet show weaker but still partially positive evidence. On Amazon, retrieved futures are only modestly closer than random futures, with $\rho(m) \approx 0.93$–0.98, suggesting that retrieval provides useful but relatively weak future information. Retweet is mixed at short horizons, where retrieved futures are

Table 10: Analog-forecasting diagnostic. We report the retrieved/random future OTD ratio $\rho(m)$ for prediction horizon $m$. Lower is better, and $\rho(m) < 1$ means the retrieved future is closer to the true future than a random training future. We also report the correlation between prefix similarity and retrieved future OTD at $m = 30$; more negative values indicate that more similar prefixes tend to have closer futures.

| Dataset | $\rho(5)$ | $\rho(10)$ | $\rho(20)$ | $\rho(30)$ | Corr. at $m = 30$ |
|---|---|---|---|---|---|
| Amazon | 0.968 | 0.976 | 0.947 | 0.934 | -0.260 |
| Retweet | 1.012 | 1.008 | 0.971 | 0.937 | -0.304 |
| StackOverflow | 0.875 | 0.809 | 0.751 | 0.763 | -0.408 |
| Taobao | 0.727 | 0.770 | 0.772 | 0.773 | -0.463 |
| Taxi | 0.890 | 0.890 | 0.844 | 0.917 | -0.103 |
| Breakfast | 0.937 | 0.914 | 0.888 | 0.829 | -0.459 |
| MultiTHUMOS | 0.968 | 1.010 | 1.016 | 1.002 | 0.003 |

comparable to random futures, but becomes more favorable at longer horizons. Taxi exhibits a different behavior: retrieved futures are better than random futures, with $\rho(m) \approx 0.84$–$0.92$, but the correlation between prefix similarity and future OTD is close to zero. This suggests that retrieval is useful on average, but the similarity score does not reliably rank which references have the closest futures. MultiTHUMOS shows the weakest analog structure under this diagnostic, with $\rho(m)$ close to or slightly above 1 for most horizons, indicating that nearest latent neighbors are often no better future templates than random references.

**Implications for ReDiTT.** These results clarify when retrieval is expected to help in asynchronous event forecasting. Retrieval is most reliable when event dynamics contain repeated, locally predictable patterns, so that similar observed prefixes provide informative future templates. This explains the strong diagnostic results on Taobao, StackOverflow, and Breakfast. In contrast, when the prefix-future relationship is noisy, ambiguous, or weakly captured by the retrieval metric, retrieved references may be imperfect or only weakly informative, as observed for MultiTHUMOS and partly for Taxi.

MultiTHUMOS is therefore best viewed as a stress test for the analog-forecasting assumption. Its weak retrieved/random OTD ratios suggest that visually or temporally similar prefixes do not always imply similar future action continuations, possibly due to greater variability in activity progression or multiple plausible futures. This highlights why ReDiTT should not be interpreted as simply copying retrieved futures. Instead, retrieved references are used as conditioning signals inside a diffusion model. When references are strong analogs, gated cross-attention allows the model to exploit them directly; when references are noisy or imperfect, diffusion-based refinement can adapt the conditioning signal to the query. Thus, retrieval provides useful non-parametric evidence when similar histories imply similar futures, while the diffusion model improves robustness when this relationship is weaker or dataset-dependent.

## C.2 Retrieval Sensitivity to Prefix-Only Neighbor Selection

A potential concern for retrieval-augmented forecasting is whether the retrieval query uses information that is unavailable at inference time. In the forecasting setting, the model observes only a prefix $x_{1:n}$ and predicts future events $x_{n+1:N}$. Therefore, a retrieved training sequence may contain its full continuation, but it should be selected using only the observed prefix. To study the practical impact of this issue without retraining all models, we perform a retrieval-only diagnostic comparing two retrieval queries: ① **Prefix retrieval**: neighbors are selected using only the observed prefix $x_{1:n}$; future target tokens are masked before computing cosine similarity. ② **Full retrieval**: neighbors are selected using the full sequence $x_{1:N}$, which acts as an oracle retrieval query.

After retrieval, we use the retrieved training sequence as a simple retrieval-only forecast and evaluate the predicted suffix with OTD. This diagnostic isolates the retrieval step: if full-sequence retrieval is much better than prefix retrieval, then future target tokens materially change neighbor selection. Conversely, if prefix retrieval is close to full retrieval, then the observed prefix already contains enough information to retrieve useful analogs, which suggests that the train-test retrieval mismatch has limited practical effect.

Table 11: Retrieval-only comparison between prefix-selected and full-sequence-selected neighbors. Values are $\text{OTD}_{\text{full}}/\text{OTD}_{\text{prefix}}$; values close to 1 indicate that prefix retrieval is comparable to oracle full-sequence retrieval.

| Dataset | Horizon 5 | Horizon 10 | Horizon 20 | Horizon 30 | Avg. |
|---|---|---|---|---|---|
| Amazon | 0.957 | 0.974 | 1.028 | 1.053 | 1.003 |
| Retweet | 0.990 | 1.002 | 1.036 | 1.052 | 1.020 |
| StackOverflow | 0.949 | 0.993 | 0.999 | 0.994 | 0.984 |
| Taobao | 0.918 | 0.893 | 0.855 | 0.809 | 0.869 |
| Taxi | 0.754 | 0.747 | 0.758 | 0.764 | 0.756 |
| Breakfast | 0.988 | 0.984 | 1.013 | 1.002 | 0.997 |
| MultiTHUMOS | 0.981 | 0.969 | 1.003 | 0.995 | 0.987 |

Table 11 reports the OTD ratio $\text{OTD}_{\text{full}}/\text{OTD}_{\text{prefix}}$ for prediction horizons 5, 10, 20, and 30. Lower OTD is better. Thus, a ratio below 1 indicates that the oracle full-sequence query improves retrieval, while a ratio near 1 indicates that prefix retrieval performs similarly to full retrieval. The results show that for five of the seven datasets, prefix retrieval is very close to full-sequence retrieval. On Amazon, Breakfast, MultiTHUMOS, Retweet, and StackOverflow, the average ratio lies between 0.984 and 1.020. In these datasets, using the future suffix in the retrieval query does not substantially improve the retrieval-only forecast. This supports the claim that the observed prefix is already sufficient to identify useful neighbors.

Taobao and Taxi are different: full-sequence retrieval gives noticeably lower OTD, especially for longer horizons. This indicates that the future suffix contains additional information that can improve oracle neighbor selection. Therefore, full-sequence retrieval should not be used as the forecasting-time query. This also motivates our evaluation protocol, where future target tokens are masked before retrieval.

Overall, this analysis helps contextualize the train-test mismatch concern. Since prefix retrieval is competitive with full-sequence retrieval on most datasets, ReDiTT can still perform well when retrieval is restricted to the information available at forecast time. This suggests that, in many event sequence datasets, the observed prefix already provides enough signal to identify useful training analogs. The stronger gap on Taobao and Taxi also indicates that the retrieval-visible token set should be stated explicitly, and that prefix-masked retrieval during training is an important direction for future work to further align training with the forecasting protocol. With further exploration on prefix based training, the Taobao and Taxi results might be also improved.

# D   Additional Ablation Results

## D.1   Retrieval-only vs. Diffusion-only Ablation.

To disentangle the contribution of retrieval from the diffusion-based conditioning architecture, we add a retrieval-only baseline alongside the diffusion-only model and full ReDiTT. The retrieval-only baseline uses the same reference bank and top-$k$ retrieval procedure as ReDiTT, but removes the diffusion model and gated cross-attention layers: after retrieving the nearest reference sequences, it directly aggregates their continuations in latent space and decodes the result with the pretrained VAE. This provides a non-parametric analog forecaster that measures how much predictive signal is already contained in the retrieved futures.

As empirically shown in Table 12, diffusion-only and retrieval-only provide complementary but comparable baselines: neither dominates the other consistently, with diffusion-only performing better on some datasets and retrieval-only performing better on others. In contrast, full ReDiTT achieves the best performance on nearly all metrics across the seven datasets, with two exceptions: retrieval-only obtains the best next-event prediction accuracy on StackOverflow, and diffusion-only achieves the best long-horizon result on Taxi. These results suggest that retrieval alone is not uniformly sufficient, and diffusion alone also does not fully explain the gains. Rather, ReDiTT's main advantage comes from combining the two: retrieved references provide useful analog futures, while the diffusion model with gated cross-attention can adapt those references to the query instead of directly copying or averaging them. The exceptions further indicate that the relative value of retrieval and generative refinement depends on the dataset, but the combined ReDiTT model is the most robust overall.

Table 12: **Next-event prediction and long-horizon prediction results** for ReDiTT, retrieval-only and diffusion-only baselines on seven benchmark datasets. RMSE is computed on the predicted inter-event time, while Accuracy ($\uparrow$) is computed on the predicted event type. Long-horizon forecasting performance is measured by OTD ($\downarrow$) at horizons $m = 5, 10, 20, 30$. We use ADiff4TPP as the unconditional diffusion-only baseline. The **best results** are highlighted in bold, and the second best results are underlined.

| | Amazon | | Retweet | | StackOverflow | | Taobao | | Taxi | | Breakfast | | MultiThumos | |
|---|---|---|---|---|---|---|---|---|---|---|---|---|---|---|
| | RMSE ($\downarrow$) | Acc% ($\uparrow$) | RMSE ($\downarrow$) | Acc% ($\uparrow$) | RMSE ($\downarrow$) | Acc% ($\uparrow$) | RMSE ($\downarrow$) | Acc% ($\uparrow$) | RMSE ($\downarrow$) | Acc% ($\uparrow$) | RMSE ($\downarrow$) | Acc% ($\uparrow$) | RMSE ($\downarrow$) | Acc% ($\uparrow$) |
| Diffusion-only | 0.413 | 33.7 | 17.480 | 60.7 | 1.524 | 34.8 | 0.140 | 57.4 | 0.309 | 85.6 | 1.360 | 8.2 | 5.981 | 16.3 |
| Retrieval-only (k=7) | 0.463 | 29.7 | 20.626 | 55.1 | **0.960** | **54.1** | 0.181 | 50.9 | 0.298 | 90.0 | 1.069 | 0.4 | 5.450 | 0.4 |
| ReDiTT (k=1) | 0.410 | 49.5 | 15.238 | 69.6 | 1.240 | 41.9 | **0.134** | 58.8 | 0.253 | 92.8 | 1.210 | 10.8 | 5.800 | 17.2 |
| **ReDiTT (k=7)** | **0.352** | **60.9** | **13.429** | **78.5** | 1.048 | 53.0 | 0.140 | 59.3 | 0.239 | 94.5 | **1.054** | **15.1** | **4.797** | **25.0** |
| | OTD ($\downarrow$) | | OTD ($\downarrow$) | | OTD ($\downarrow$) | | OTD ($\downarrow$) | | OTD ($\downarrow$) | | OTD ($\downarrow$) | | OTD ($\downarrow$) | |
| Diffusion-only | 6.2 / 12.4 / 24.7 / 32.9 | | 9.1 / 17.7 / 28.0 / 31.7 | | 6.5 / 12.0 / 22.3 / 30.1 | | 4.9 / 9.9 / 20.4 / 31.1 | | **2.4 / 4.0 / 6.8 / 9.4** | | 8.8 / 17.1 / 24.3 / 28.9 | | 8.2 / 13.6 / 21.1 / 20.2 | |
| Retrieval-only(k=7) | 6.9 / 13.1 / 22.6 / 27.4 | | 9.4 / 18.0 / 27.4 / 29.9 | | 6.1 / 11.5 / 20.0 / 28.5 | | 5.3 / 10.1 / 19.2 / 28.8 | | 2.8 / 4.3 / 6.7 / 9.1 | | 9.4 / 18.0 / 27.6 / 33.6 | | 9.4 / 17.2 / 26.4 / 27.8 | |
| ReDiTT (k=1) | 5.9 / 11.5 / 21.3 / 27.8 | | 9.1 / 17.3 / 26.9 / 30.1 | | 6.1 / 10.8 / 19.4 / 27.4 | | 4.8 / 9.8 / 19.8 / 30.7 | | 2.6 / 4.4 / 7.9 / 12.7 | | 8.7 / 16.1 / 22.1 / 27.0 | | 8.2 / 14.3 / 20.6 / 19.6 | |
| **ReDiTT (k=7)** | **5.7 / 10.7 / 19.7 / 25.8** | | **9.1 / 17.2 / 26.5 / 29.6** | | **6.0 / 10.7 / 18.9 / 27.4** | | **4.6 / 9.4 / 19.4 / 29.5** | | 3.5 / 5.0 / 9.2 / 17.6 | | **8.2 / 15.4 / 21.6 / 26.4** | | **8.2 / 14.1 / 20.6 / 19.6** | |

## D.2 Latent vs. Raw-space Retrieval Ablation.

We further evaluate a raw-space retrieval ablation to test whether ReDiTT's performance depends specifically on retrieving neighbors in the VAE latent space. In this variant, we keep the trained ReDiTT model, VAE, reference bank size, and diffusion-based conditioning architecture fixed, and change only the neighbor-selection rule. Instead of computing similarity between latent event representations, we retrieve references using normalized raw event tuples, combining log-scaled inter-event time distance with event-type matching over the observed prefix. The retrieved references are still provided to the diffusion model as latent sequences, so the ablation isolates the effect of the retrieval space while preserving the rest of the pipeline. As listed in Table 13, latent-space retrieval performs better on most datasets and metrics, but the gap is generally small: raw-space retrieval remains competitive and often produces similar results. This suggests that retrieval-augmented conditioning is itself a robust source of improvement, while latent-space retrieval provides a modest but consistent advantage and is better aligned with ReDiTT's latent diffusion target. It also avoids the need to hand-design raw-space similarities over mixed continuous-discrete, padded asynchronous sequences.

Table 13: **Next-event prediction and long-horizon prediction results** for ReDiTT, and Raw-space Retrieval baseline on seven benchmark datasets for $k = 7$. RMSE is computed on the predicted inter-event time, while Accuracy ($\uparrow$) is computed on the predicted event type. Long-horizon forecasting performance is measured by OTD ($\downarrow$) at horizons $m = 5, 10, 20, 30$. The **best results** are highlighted in bold, and the second best results are underlined.

| | Amazon | | Retweet | | StackOverflow | | Taobao | | Taxi | | Breakfast | | MultiThumos | |
|---|---|---|---|---|---|---|---|---|---|---|---|---|---|---|
| | RMSE ($\downarrow$) | Acc% ($\uparrow$) | RMSE ($\downarrow$) | Acc% ($\uparrow$) | RMSE ($\downarrow$) | Acc% ($\uparrow$) | RMSE ($\downarrow$) | Acc% ($\uparrow$) | RMSE ($\downarrow$) | Acc% ($\uparrow$) | RMSE ($\downarrow$) | Acc% ($\uparrow$) | RMSE ($\downarrow$) | Acc% ($\uparrow$) |
| Latent-space | 0.352 | 60.9 | 13.429 | 78.5 | 1.048 | 53.0 | 0.140 | 59.3 | 0.239 | 94.5 | 1.054 | 15.1 | 4.797 | 25.0 |
| Raw-space | 0.280 | 47.6 | 14.348 | 80.2 | 0.997 | 47.2 | 0.130 | 56.1 | 0.213 | 92.1 | 1.094 | 9.9 | 5.052 | 20.4 |
| | OTD ($\downarrow$) | | OTD ($\downarrow$) | | OTD ($\downarrow$) | | OTD ($\downarrow$) | | OTD ($\downarrow$) | | OTD ($\downarrow$) | | OTD ($\downarrow$) | |
| Latent-space | 5.7 / 10.7 / 19.7 / 25.8 | | 9.1 / 17.2 / 26.5 / 29.6 | | 6.0 / 10.7 / 18.9 / 27.4 | | 4.6 / 9.4 / 19.4 / 29.5 | | 3.5 / 5.0 / 9.2 / 17.6 | | 8.2 / 15.4 / 21.6 / 26.4 | | 8.2 / 14.1 / 20.6 / 19.6 | |
| Raw-space | 5.8 / 10.9 / 20.0 / 26.1 | | 9.1 / 17.2 / 26.6 / 29.5 | | 6.1 / 10.9 / 19.6 / 27.8 | | 4.9 / 10.0 / 19.8 / 30.4 | | 3.5 / 4.9 / 8.8 / 17.3 | | 8.8 / 16.4 / 22.9 / 28.1 | | 8.2 / 14.0 / 20.5 / 19.4 | |

