# OpenReview forum: "ReDiTT: Retrieval Augmented Conditional Diffusion Transformers for Asynchronous Time Series"
_TMLR — Accepted by TMLR_

### Review · Reviewer_nEGP · 2026-05-20

**Summary Of Contributions:**

The paper introduces ReDiTT, a retrieval-augmented latent diffusion transformer for asynchronous time series / temporal point processes (TPPs). The method is built on top of the recently proposed ADiff4TPP, inheriting its β-VAE latent encoding of (inter-event time, event type) pairs and its asynchronous matrix-valued noise schedule for Conditional Flow Matching. This work introduces:

1) A masked token-level cosine similarity top-k retrieval over a precomputed latent token bank built from the training set
2) A softmax-weighted average pooling aggregation of the k retrieved latents into a single reference of the same shape as the query.
3) A gated multi-head cross-attention module inserted after self-attention in each DiT block, which lets each query latent token attend to positions of the aggregated reference, replacing or complementing the AdaLN conditioning used in standard DiTs.
4) Empirical evaluation on seven TPP benchmarks (Amazon, Retweet, StackOverflow, Taobao, Taxi, Breakfast, MultiThumos) for next-event prediction (RMSE, type accuracy) and long-horizon prediction (OTD at horizons 5/10/20/30), plus ablations over the conditioning architecture (cross-attn vs AdaLN), the aggregation operator (pooling vs concatenation), the retrieval size k and the per-modality (time-only / type-only) contribution.

Strengths:
- Retrieval augmentation is novel in the context of marked TPPs/asynchronous event streams. Earlier retrieval-augmented diffusion work (RATD, TS-RAG, etc.) targets regularly sampled segments and does not handle event-wise (time, mark) structure or variable-length padded sequences in a principled way
- The masked cosine similarity that ignores padded positions is functional, given the heterogeneity in sequence lengths, and is described clearly
- The empirical improvements are large and consistent on Breakfast and MultiThumos, which are exactly the regimes (large mark vocabulary, few training sequences) where parametric TPPs are known to underperform
- The ablations on cross-attn vs AdaLN are informative, and the explanation  that AdaLN is a poor fit when the condition is high-entropy and event-dependent rather than a single global discrete label  is reasonable and well-argued

Weaknesses:
- The claim of a “novel conditioning approach” in Contribution 2 should be scaled down, since gated cross-attention over retrieved memory is standard practice (e.g., RETRO, RATD). The novelty lies instead in combining it with the asynchronous flow-matching backbone and masked token-wise similarity.
- The novelty over ADiff4TPP is narrower than the framing suggests: VAE encoder, asynchronous matrix-valued interpolation A(s), the schedule in Eq. (16), the timestep embedding, the diagonal-driven conditioning and even most hyperparameters are directly inherited.
- The retrieval cost (memory of the bank, indexing strategy) is only briefly addressed in Limitations; in particular, no information is given on how retrieval scales with the size of the training bank or what happens on larger datasets.

**Audience:**

Yes

**Audience Explanation:**

TMLR has a sustained readership in temporal point processes, diffusion / flow-matching generative models, and retrieval-augmented generation. Each of these communities would find something of interest: Surely, the findings are of interest to some subset of the community.

**Broader Impact Concerns:**

The paper lacks a Broader Impact statement. This is not a barrier to acceptance, but a brief note on misuse, applications and over-trust would be appropriate.

**Claims And Evidence:**

Yes

**Claims Explanation:**

The main empirical claims that retrieval-augmented conditioning improves both next-event and long-horizon prediction in most datasets, and that cross-attention conditioning outperforms AdaLN for this signal, are reasonably supported by Tables 1-3. The pattern across Breakfast and MultiThumos (the regime the authors highlight) is consistent, and the gains there are large enough to be convincing.

However, some aspects weaken the evidence:
- Missing baselines: For Breakfast and MultiThumos, it is not clear why DTPP and HYPRO baselines are not reported: an explicit note, as for Add&Thin, for each missing cell would strengthen credibility.

**Requested Changes:**

1) Behaviour on OOD prefixes. The paper retrieves from the training set at inference time, but does not discuss what happens when the test prefix differs substantially from anything in the training set. In such cases, the top-k retrievals will still return something, but those neighbours may be semantically irrelevant and risk misleading the model rather than guiding it. A held-out OOD experiment, or at least a qualitative analysis of retrieval similarity scores at test time, would clarify how robust the approach is in this regime.

2) Hyperparameter sensitivity. DiT depth 7, hidden 1152, 16 heads are identical across all seven datasets. Add a justification or a small sensitivity study.

3) Training-iteration counts differ across configurations, and it would be nicer to have a useful and clearer investigation into this

3) Typos and minor issues:
   - "long horizontal forecasting" (Sec. 1)  "long-horizon forecasting".
   - Eq. (4): "n < i ≤ m" should be "n < i ≤ n+m".

---

> ### Author Response · Authors · 2026-06-09
> **Rebuttal Reply**
>
> We thank Reviewer nEGP for the careful and constructive review. We appreciate the clear summary of the paper, the recognition that retrieval augmentation is useful for marked asynchronous event streams, and the helpful suggestions on sharpening the framing and improving the empirical discussion. We have revised our paper (marked in red) to better support our claims.
>
> - **Behaviour on OOD prefixes**. We agreed that this is important in retrieval-based method, we thank the reviewer for bringing this up. We have add an analysis of test-time retrieval similarity scores and discuss failure modes for low-similarity prefixes in **Appendix C.1**. We found that MultiTHUMOS is therefore viewed as a stress test/nearly OOD example for the analog-forecasting assumption, since it shows nearest latent neighbors are often no better future templates than random references. However, ReDiTT gains much for MultiThumos compared to other baselines, which indicates that when references are noisy or imperfect, diffusion-based refinement can adapt the conditioning signal to the query.
>
> - **Hyperparameter sensitivity**. We use the same DiT architecture across all datasets to keep the experimental comparison controlled. In particular, we follow ADiff4TPP and use the same DiT configuration, rather than tuning the backbone separately for each dataset. This design choice allows us to isolate the gains from the proposed retrieval-augmented conditioning mechanism and the diffusion-based temporal modeling component (details are added in Appendix D.1), instead of conflating them with dataset-specific architecture tuning. While further tuning of DiT depth, hidden dimension, or number of attention heads may improve absolute performance on individual datasets, due to limit of time, we leave such dataset-specific optimization to future work and focus here on evaluating the effect of retrieval under a fixed backbone.
>
> - **Training-iteration**. We thank the reviewer for pointing this out. Training iteration is indeed one of the main hyperparameters we tuned, and we have clarified the checkpoint-selection protocol in the revision in **Appendix B.4 and Table 5** as an example for MultiThumos. For each configuration, we trained the model for up to 500k iterations, saved checkpoints every 50k iterations, and evaluated each checkpoint using all three reported criteria: next-event time error, next-event type accuracy, and long-horizon OTD. We then reported the checkpoint that gave strong and balanced performance across these metrics, rather than selecting using only a single metric.
>
> - **Typos and minor issues**. We thank the reviewer for carefully read our submission and point this out. We have fixed them.
>
> - **Broader Impact statement**. We have included an broader impact statement in the **Appendix A**.
>
> - **Retrieval cost**. We have added a discussion of memory cost in **Appendix B.5**. We hope this discussion help practitioners assess deployment viability.
>
> - **Missing baselines**. We have filled in the missing values in Table 1. Since
> Add \&Thin only predicts the inter-event time instead of event type, we only reported the RMSE in Table 1. Consequently, we did not compute the OTD since event types are missing.
>
> Overall, we thank the reviewer for the thoughtful and constructive feedback. The suggested revisions helped us sharpen the positioning of ReDiTT, clarify the role of retrieval in the model’s improvements, and provide a more transparent discussion of experimental choices and practical limitations. We believe these changes make the paper clearer, more precise, and better supported.

---

### Review · Reviewer_J6Qq · 2026-05-28

**Summary Of Contributions:**

This paper proposes ReDiTT, the first retrieval-augmented latent diffusion framework for asynchronous time series forecasting, which retrieves top-k similar trajectories from a VAE latent token bank and uses them as conditioning for a diffusion transformer.
To avoid the brittleness of retrieval in raw event space (scale mismatch, padding, alignment), the authors perform retrieval in the VAE latent space using a padding-aware, token-wise masked cosine similarity, and exclude the query itself during training to prevent trivial conditioning.
The authors extend the asynchronous matrix-valued conditional flow matching objective by conditioning the learned vector field on the retrieved references, biasing the dynamics toward trajectory-consistent solutions.
They argue that DiT's standard AdaLN-style global conditioning is too weak for asynchronous sequences and instead inject retrieved references as token-level external memory via a sigmoid-gated cross-attention module whose gate starts weak and strengthens only when retrieval helps.
Experiments on seven datasets claim SOTA on both next-event and long-horizon prediction against RNN, attention, and diffusion-based baselines, with the largest gains reported in the long-sequence and limited-training-data regimes.

**Audience:**

Yes

**Audience Explanation:**

The paper sits at the intersection of three active TMLR-relevant areas: (1) neural temporal point processes and asynchronous event-sequence modeling, (2) diffusion / flow-matching generative models for sequential data, and (3) retrieval-augmented generation. Each of these communities is well represented in the TMLR readership, and the paper's central design, combining latent-space retrieval with a conditional flow-matching DiT for marked event sequences, is novel as far as I can tell. The empirical results are sufficiently strong across seven datasets to be of interest to practitioners working on long-horizon event forecasting (healthcare, finance, user behavior). I therefore believe a sufficient subset of the TMLR audience will find the work interesting and useful.

**Claims And Evidence:**

Yes

**Claims Explanation:**

The paper's central claims are (1) retrieval-augmented conditioning improves asynchronous time series forecasting, (2) the gains are especially pronounced in long-horizon prediction and in regimes with large event vocabularies and limited training data, and (3) cross-attention is a more suitable conditioning mechanism than AdaLN-style global conditioning for this setting. The empirical evidence broadly supports these claims. Main results across seven benchmarks (Table 1) are reported with three random seeds and standard deviations, and ReDiTT achieves the best or second-best RMSE, type accuracy, and OTD on the large majority of dataset–metric pairs, with the largest gains on Breakfast and MultiThumos as predicted by the high-cardinality / low-data argument. Long-horizon improvements grow with horizon size on Amazon and are consistent on Retweet and StackOverflow, lending direct support to the long-horizon claim.

That said, several claims are only partially supported by the evidence shown:

- **"adaLN is insufficient for asynchronous time series."** Table 2 actually shows that adaLN matches or outperforms cross-attention on next-event type accuracy across all five tested datasets (notably 51.3 vs. 41.9 on StackOverflow, 60.5 vs. 58.8 on Taobao, 94.0 vs. 92.8 on Taxi). Cross-attention wins on OTD (long-horizon) but loses on short-horizon classification. The paper presents this as a clear win for cross-attention, but the trade-off is more nuanced than the text suggests.
- **Aggregation and $k$ ablations** (Table 3) are reported on StackOverflow only, and **time-vs-type ablation** (Table 4) on Taobao only. The design decisions (average pooling, $k=7$, both modalities) generalize to the main experiments without being verified to generalize across datasets.
- **Retrieval in latent vs. raw space** is presented as a key design choice (§4.1), but the paper does not include a controlled comparison showing that retrieval in raw event space actually underperforms, the motivation is argued conceptually but not measured.
- **Retrieval contribution vs. architectural contribution.** No retrieval-only non-parametric baseline is reported (e.g., averaging the retrieved continuations directly in latent space without diffusion or cross-attention), so it is hard to attribute gains specifically to retrieval as opposed to the diffusion + gated cross-attention machinery built on top.

None of these gaps undermines the headline claim that retrieval helps in this setting, but they leave some of the finer claims under-evidenced.

**Requested Changes:**

**Major**

1. **Retrieval-only baseline.** Add a simple non-parametric forecaster that retrieves the same top-$k$ references and aggregates their continuations directly in latent space (e.g., weighted pooling followed by decoding) without diffusion or cross-attention. This is needed to isolate how much of the empirical gain comes from retrieval itself versus the diffusion + gated cross-attention machinery built on top, which is currently the most ambiguous attribution in the paper.

2. **Latent vs. raw-space retrieval ablation.** §4.1 motivates latent-space retrieval as a key design choice, but this is not directly measured. Please add a controlled comparison where retrieval is performed in raw event space (e.g., on padded $(t,e)$ tuples with a comparable similarity metric) and the rest of the pipeline is held fixed, to verify that latent-space retrieval is empirically superior.

3. **Generalize the ablations beyond a single dataset.** The aggregation/$k$ ablation (Table 3) is run only on StackOverflow, and the time-vs-type ablation (Table 4) only on Taobao. Since these results justify design choices used everywhere, please repeat them on at least 2–3 additional datasets to show that the chosen settings ($k=7$, average pooling, both modalities) are robust.

4. **More accurate framing of the adaLN vs. cross-attention comparison.** Table 2 shows that adaLN actually matches or beats cross-attention on next-event type accuracy on all five tested datasets, while cross-attention wins on long-horizon OTD. Please revise the surrounding discussion (§5.3 "Different Conditioning Architecture") to reflect this trade-off explicitly rather than presenting cross-attention as uniformly superior.

**Minor**

1. **Train/inference asymmetry in retrieval.** During training the closest reference (the query itself) is excluded, but at inference the top-1 is included. Please report whether including the top-1 at inference empirically helps, and whether the asymmetry causes any measurable distributional gap.

2. **Conceptual discussion of when retrieval helps for time series.** Unlike NLP or image retrieval, time-series retrieval implicitly relies on the assumption that "futures of similar pasts" are good templates for the query's future — a form of analog forecasting. A short discussion of when this assumption holds (stationary, locally smooth processes) versus where it can break (highly chaotic or non-stationary processes) would help readers calibrate when ReDiTT is expected to be most useful, and would partially explain the Taxi outlier behavior in Table 1.

3. **More rigorous overhead reporting.** Appendix B.4 reports inference latency for next-event prediction (2.1s vs. 2.0s baseline) but does not quantify the memory cost of the reference bank or how cost scales with bank size. Even a brief discussion would help practitioners assess deployment viability.

---

> ### Author Response · Authors · 2026-06-09
> **Rebuttal Reply**
>
> We thank the reviewer J6Qq for the careful and constructive evaluation. We appreciate the thoughtful assessment of the paper and the helpful suggestions for improving the framing, empirical presentation, and discussion of limitations. We have revised the manuscript (changes marked in red), to make the claims clearer and better supported.
>
> - **Retrieval-only baseline**. To isolate the contributsion of retrieval from diffusion and the gated cross-attention architecture, we added a retrieval-only baseline in **Appendix Section D.1**. This baseline uses the same reference banck and topk retrieval procedure as ReDiTT, but removes the diffusion model and cross-attention layers: the retrieved continuations are directly aggregated in latent space and decoded with the pretrained VAE. We compare diffusion-only, retrieval-only, and full ReDiTT. Empirically, diffusion-only and retrieval-only are comparable baselines: neither consistently dominates the other, with each performing better on different datasets. In contrast, full ReDiTT achieves the best performance on nearly all metrics across the seven datasets.
>
> - **Raw-space ablation**. We added a raw-space retrieval ablation in **Appendix Section D.2**, to test whether ReDiTT’s gains depend specifically on retrieving neighbors in the VAE latent space. Empirically, latent-space retrieval performs better on most datasets and metrics, but the gap is generally small: raw-space retrieval remains competitive and often produces similar results. We therefore revised the manuscript to frame it as a representation-aligned design choice that is natural for latent diffusion and avoids hand-designing raw-space similarities over mixed continuous-discrete, padded asynchronous sequences.
>
> - **Generalize ablations**. We expanded the ablation studies beyond the original single-dataset settings, in both **Table 3 and Table 4**. We repeated the aggregation and $k$ ablations on Breakfast and MultiTHUMOS in addition to StackOverflow, and repeated the time-only/type-only ablations on Breakfast and MultiTHUMOS in addition to Taobao. These experiments provide broader evidence that the design choices used in the main model are not specific to one dataset.
>
> - **Framing of adaLN**. We revised the discussion of adaLN versus cross-attention in **section 5.3 Different Conditioning Architecture**. We explicitly discuss the trade-off shown in the results: adaLN can match or outperform cross-attention on next-event type accuracy, while cross-attention is more effective for long-horizon OTD. We interpret this as evidence that global conditioning can be sufficient for short-horizon classification, whereas token-level cross-attention is more useful when the model must generate coherent multi-event futures from retrieved references.
>
> - **Train/inference assymmetry**. The asymmetry is intentional and follows from the different retrieval pools used during training and inference. During training, the query sequence itself belongs to the training reference bank. If we include the nearest neighbor, the model can retrieve an exact duplicate of the input sequence, which would create a degenerate shortcut rather than useful retrieval conditioning. We therefore exclude the closest reference during training. At inference, however, the query comes from the validation/test set while the reference bank contains only training sequences, so the top-1 retrieved reference cannot be the query itself. Thus, including the top-1 neighbor at inference does not introduce the same leakage issue. We have clarified this distinction in the manuscript in **section 4.1**.
>
> - **Conceptual discussion of when retrieval helps for time series**. We agree that retrieval for time-series/event-sequence forecasting relies on an analog-forecasting assumption: similar observed histories should provide useful templates for future continuations. We have added a new analysis discussing and empirically diagnosing this assumption in **Appendix C.1**. We have added this discussion to the appendix to clarify when ReDiTT is expected to be most useful: retrieval helps most when the process is stationary or locally smooth enough that similar pasts imply similar futures, while in highly stochastic, non-stationary, or multi-modal settings retrieved references may be imperfect. This also clarifies why ReDiTT uses retrieval as conditioning for a diffusion model rather than directly copying retrieved futures.
>
> - **More rigorous overhead reporting**. We have added a discussion of memory cost in **Appendix section B.5**. We hope this discussion help practitioners assess deployment viability.
>
> Overall, we thank the reviewer for the constructive feedback. The requested analyses helped us clarify the attribution of ReDiTT’s gains, better qualify when retrieval is expected to help, and make the empirical and deployment discussion more rigorous. We believe these additions make the submission clearer, more balanced, and better supported experimentally.

---

### Review · Reviewer_MXse · 2026-05-28

**Summary Of Contributions:**

The paper proposes ReDiTT, a retrieval-augmented conditional diffusion transformer for asynchronous time-series / temporal point-process forecasting. The method encodes event sequences into a VAE latent space, retrieves top-k similar latent sequences from a training memory bank, and injects the retrieved references into a DiT-style flow-matching model via cross-attention. The paper evaluates on seven datasets for next-event prediction and long-horizon forecasting, reporting strong gains over neural TPP, diffusion, and other baselines.

Strengths:
- The problem is important and timely. Retrieval-augmented generation is well motivated for long-horizon event forecasting, where purely parametric diffusion models may drift toward generic futures.
- The proposed architecture is intuitive: retrieving event-level latent references and injecting them through cross-attention is a reasonable design for conditioning a diffusion transformer on trajectory-specific information.
- The empirical section is broad. The paper compares against many temporal point-process baselines, including RMTPP, NHP, SAHP, THP, AttNHP, DTPP, Add&Thin, HYPRO, IFTPP, and ADiff4TPP, and evaluates both next-event and long-horizon metrics across seven datasets.
- The ablations are useful. In particular, the comparisons between cross-attention and adaLN, average pooling and concatenation, and different retrieval sizes help justify several design choices.

Weaknesses:
- The main concern is potential future-information leakage or train-test mismatch in retrieval. The method description defines retrieval using latent representations of the full sequence, but the forecasting task should retrieve using only the observed prefix. During training, if the query used for retrieval includes future target tokens, then the retrieved neighbors may be selected using information unavailable at inference. Even if this is not what the implementation does, the paper needs to state this clearly and provide an algorithm showing exactly which tokens are visible to retrieval during training and evaluation.
- The novelty is somewhat incremental. ReDiTT combines latent diffusion for TPPs, retrieval augmentation, and cross-attention conditioning. Each component is individually well established. The paper’s contribution would be stronger if it more sharply isolated what is unique to asynchronous marked event streams beyond adapting retrieval-conditioned diffusion to this domain.

**Audience:**

Yes

**Audience Explanation:**

The topic is highly related to machine learning.

**Claims And Evidence:**

Yes

**Claims Explanation:**

The experimental part makes sense, with a clear problem setup, evaluation setup, and baseline selection.

**Requested Changes:**

1. Do the authors can explain does the model have the information leakage issue. If not, how the model mitigates the leakage?
2. Do the authors can provide more clear distinction between the method and other approaches to emphasize the strengths.

---

> ### Author Response · Authors · 2026-06-09
> **Rebuttal reply**
>
> We thank the reviewer MXse for the thoughtful assessment and for pointing out that the paper should more clearly distinguish ReDiTT’s contribution from the established ingredients it builds upon. We have revised the paper (changes mark in red) to make the claims clearer.
>
> - **Train-test mismatch**. We thank the reviewer MXse for raising this point. We clarify that this is not test-set leakage: ReDiTT never uses test-set future information to build the retrieval database or to select neighbors at evaluation time. The retrieval database is constructed only from the training set. During forecasting evaluation, all target/future events in the test sequence are masked before cosine similarity is computed, so neighbor selection uses only the observed prefix available at prediction time. To better understand the impact of using prefix-only retrieval, we conducted an additional retrieval-only analysis, reported in **Appendix C.2**. We compare prefix retrieval against oracle full-sequence retrieval, where the latter uses the complete sequence to select neighbors. On five of seven datasets, prefix retrieval is competitive with full-sequence retrieval. These results indicate that the observed prefix is generally sufficient to retrieve useful training analogs, helping explain why ReDiTT remains effective under the forecasting-time visibility constraint. We also observe larger differences on Taobao and Taxi, suggesting that prefix-masked retrieval during training is a useful direction for future work to further align training with the evaluation protocol. Overall, the additional analysis supports that ReDiTT’s retrieval mechanism remains meaningful when restricted to information available at forecast time.
>
> - **Method Isolation**. We thank the reviewer for this important question. Our contribution claims that the model must jointly handle continuous event times, discrete marks, variable sequence lengths, and variable forecasting horizons. To better isolate the contribution of each component, we have added additional comparisons in **Appendix D.1**, including a retrieval-only baseline and a diffusion-only baseline. These results show that retrieval alone is insufficient and that diffusion without retrieval also underperforms ReDiTT, indicating that the gains come from combining retrieval with the generative diffusion model rather than from either component alone. We also clarify the role of cross-attention conditioning in **Section 5.3**. There, we compare different conditioning architectures and include an unconditional diffusion baseline. The results show that explicitly conditioning the diffusion backbone on retrieved references through cross-attention is more effective than removing retrieval or using weaker conditioning alternatives to have good performances across all of the metrics. Overall, these additional analyses help to clarify the empirical contribution: ReDiTT shows that retrieval-conditioned diffusion with cross-attention is an effective architecture for asynchronous marked event forecasting.
>
> Overall, we thank the reviewers for these constructive comments. We have revised the paper to more clearly specify the information available to retrieval during forecasting and to better isolate ReDiTT’s empirical contributions. We show that prefix retrieval remains competitive with oracle full-sequence retrieval on most datasets. We also add retrieval-only and diffusion-only baselines in Appendix D.1 and clarify in Section 5.3 that cross-attention-based retrieval conditioning improves over unconditional diffusion and weaker conditioning alternatives.

---

### Decision · Action_Editor_6uwA · 2026-07-07

**Recommendation:** Accept as is

**Audience:**

Yes

**Audience Explanation:**

Time series modeling is a core interest in machine learning, especially when paired with diffusion models and transformers.

**Claims And Evidence:**

Yes

**Claims Explanation:**

The authors introduce a diffusion-based time series model, they present numerous empirical results that support the main contribution of the paper (especially after the rebuttal phase).

---

> ### Author Response · Authors · 2026-07-13
> **Upload camera ready**
>
> Dear AE,
>
> We have uploaded our camera-ready version based on your final comments. Thank you for smoothly handling our submission!
>
> Best wishes,
>
> Authors